# Impact of faults on the remote stress state

Karsten Reiter[1], Oliver Heidbach[2,3], and Moritz. O. Ziegler[2,4]

[1]Institute of Applied Geosciences, TU Darmstadt, 64287 Darmstadt, Germany
[2]Helmholtz Centre Potsdam, GFZ German Research Centre for Geosciences, 14473 Potsdam, Germany
[3]Institute of Applied Geosciences, TU Berlin, 10587 Berlin, Germany
[4] TU Munich, 80333 Munich, Germany

**Correspondence:** Karsten Reiter (reiter@geo.tu-darmstadt.de)

**Abstract.**

The impact of faults on the contemporary stress field in the upper crust has been discussed in various studies. Data and models clearly show that there is an effect, but so far, a systematic study quantifying the impact as a function of distance from the fault is lacking. In the absence of data, here we use a series of generic 3-D-models to investigate which component of the stress tensor is affected at which distance from the fault. Our study concentrates on the far-field, located hundreds of metres from the fault zone. The models assess various techniques to represent faults, different material properties, different boundary conditions, variable orientation, and the fault's size. The study findings indicate that most of the factors tested do not have an influence on either the stress tensor orientation or principal stress magnitudes in the far field beyond 1000 m from the fault. Only in the case of oblique faults with low static friction coefficient of $\mu = 0.1$, noteworthy stress perturbations can be seen up to 2000 m from the fault. However, the changes that we detected are generally small an in the order of lateral stress variability due to rock property variability. Furthermore, only in the first 100's of meters distance to the fault variations are large enough to be theoretically detected by borehole based stress data when considering their inherent uncertainties. This finding agrees with robust stress magnitude measurements and stress orientation data. Thus, in areas where high-quality and high-resolution data show a gradual and continuous stress tensor rotations of $> 20°$ are observed over lateral spatial scales of 10 km or more we infer that these rotations cannot be attributed to faults. We hypothesise that most stress orientation changes attributed to faults may originate from different sources such as density and strength contrasts.

## 1 Introduction

The crustal stress field is a key driver of geodynamic processes such as the earthquake cycle (Heidbach and Ben-Avraham, 2007; Wang et al., 2015; Hardebeck and Okada, 2018; Brodsky et al., 2020) and is of great importance for the safe exploitation of georeservoirs and storage of energy or waste in the subsurface (Fuchs and Müller, 2001; Zoback, 2010; Smart et al., 2014).

In this context the interaction between the stress field in the Earth's upper crust and pre-existing faults is a crucial issue (Yale, 2003; Schoenball and Davatzes, 2017; Blöcher et al., 2018; Kruszewski et al., 2022; Röckel et al., 2022; Li et al., 2023).

For practical applications it is important, to understand and to quantify on which spatial scale the fault changes the stress state. Exemplified on the site selection process for a deep geological repository for high-level radioactive waste, the interest is to know the distance to a fault at which no significant changes of the stress components occur in order to build the repository in a rock volume with homogeneous stress field conditions. In contrast to this, deep geothermal exploration targets faults or fault networks since they provide higher permeabilities compared to the rock matrix. Thus, the changes of the stresses in the near-field of the fault and in its core or fracture network is of key interest to assess its dilation tendency (Moeck and Backers, 2011; Seithel et al., 2019; Ferrill et al., 2020). Stress perturbations are also significant for evaluating secondary fracturing near faults and its associated permeability, which encompasses joint orientation, secondary faulting, and bed-parallel slip (e.g. Kattenhorn et al., 2000; Maerten et al., 2002; Delogkos et al., 2022).

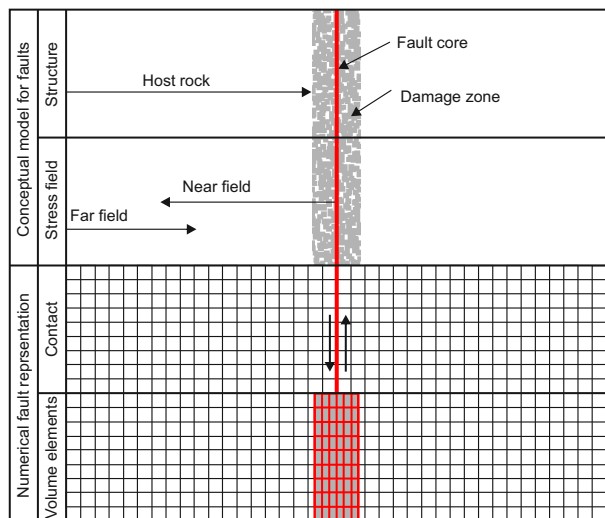

**Figure 1.** The general structure of a fault is described by the fault core, the damage zone, and the host rock (e.g., Caine et al., 1996; Faulkner et al., 2003). The purpose of this study is not to explore the effect of a fault on the stress state in the near field. This would include the fault core, the damage zone, and the neighbouring host rock. The study is focused on the far-field stress state, which is located several tens or hundreds of metres away from the fault and can extend up to a few kilometres at most. Numerical models typically employ one or a combination of two principle technical fault representations. Contact surfaces are a discontinuity within the mesh, where relative offset of the mesh is allowed, mainly depending on the friction. The second method use a continuous mesh with elements having a lower stiffness or a failure criterion which results in a distributed deformation within the defined fault representation elements.

One of the key questions is, on what spatial scale faults change the stress field and to quantify which stress components are affected. The only component of the 3-D stress tensor that is systematically compiled is the orientation of maximum horizontal stress ($S_{Hmax}$, Heidbach et al., 2004, 2018). Areas with high data density revealed that the $S_{Hmax}$ orientation can rotate significantly on scales from 10's to 100's of kilometres (Tingay et al., 2006; Heidbach et al., 2007; Rajabi et al., 2017b;

Heidbach et al., 2018; Lund Snee and Zoback, 2020). The cause of this spatial variability has been investigated with generic geomechanical-numerical and analytical modelling (e.g. Sonder, 1990; Reiter, 2021). These studies show that stiffness, strength and density contrasts are certainly a key driver of spatial distributed changes of the $S_{Hmax}$ orientation.

**Table 1.** List of some studies, exemplifying the use of either a continuous or discontinuous mesh for fault representation (Fig. 1). Discontinuities are represented by contact elements or comparable methods (Contact). Another method of modelling faults is utilising a continuous mesh that possesses a material definition slightly or significantly weaker (elastic, plastic or viscous). These are 2-D elements within a 2-D Mesh or 3-D Elements in a 3-D mesh (Volume). Many models apply the Finite Element Method (FEM), others use the Finite Difference Method (FDM), Finite Volume Method (FVM), the Discrete Element Method (DEM). (The list does not claim to be complete.)

| | Authors | Contact | Volume |
|---|---|---|---|
| Finite Element Method | Tommasi et al. (1995) | - | x |
| | Buchmann and Connolly (2007) | x | - |
| | Xing et al. (2007) | x | - |
| | Hergert et al. (2011) | x | - |
| | Reiter and Heidbach (2014) | x | - |
| | Pereira et al. (2014) | - | x |
| | Hergert et al. (2015) | x | - |
| | Franceschini et al. (2016) | x | - |
| | Zhang et al. (2016) | - | x |
| | Meier et al. (2017) | - | x |
| | Schuite et al. (2017) | - | x |
| | Treffeisen and Henk (2020b) | x | x |
| | Reiter (2021) | x | - |
| other methods | Homberg et al. (1997) | x | - |
| | Sánchez D. et al. (1999) | x | - |
| | Maerten et al. (2002) | x | - |
| | McLellan et al. (2004) | - | x |
| | Camac and Hunt (2009) | x | - |
| | Cappa (2009) | - | x |
| | Cappa and Rutqvist (2011) | x | - |

Besides these findings, it was also hypothesised that active faults can cause rotations or magnitude variations as well (Dart and Swolfs, 1992; Yale, 2003; Faulkner et al., 2006; Konstantinovskaia et al., 2012; Schoenball et al., 2018; Li et al., 2023). This is confirmed on borehole scale since logging data show stress rotations on the meter scale by means of abrupt changes in the orientation of borehole breakouts and drilling induced tensile fractures (e.g., Barton and Zoback, 1994; Zoback et al., 2011; Rajabi et al., 2017c). It clearly showed that there are indeed stress rotations on scales of one to several hundreds of
meters occur, due to faults and that the amount of rotation changes with distance to the fault core (Hickman and Zoback, 2004).

Significant variation of stress magnitudes in the vicinity of faults has been reported for China and Scandinavia (Stephansson and Ångman, 1986; Li et al., 2023), but from these studies it is not clear which stress tensor component is affected as a function of distance to the fault. Furthermore, the mix of different methods that are used to estimate stress parameter from very shallow locations near surface as well as the lack of a rigorous uncertainty assessment makes it difficult to assess whether the observed changes are significant and if they can be exclusively attributed to the nearby fault.

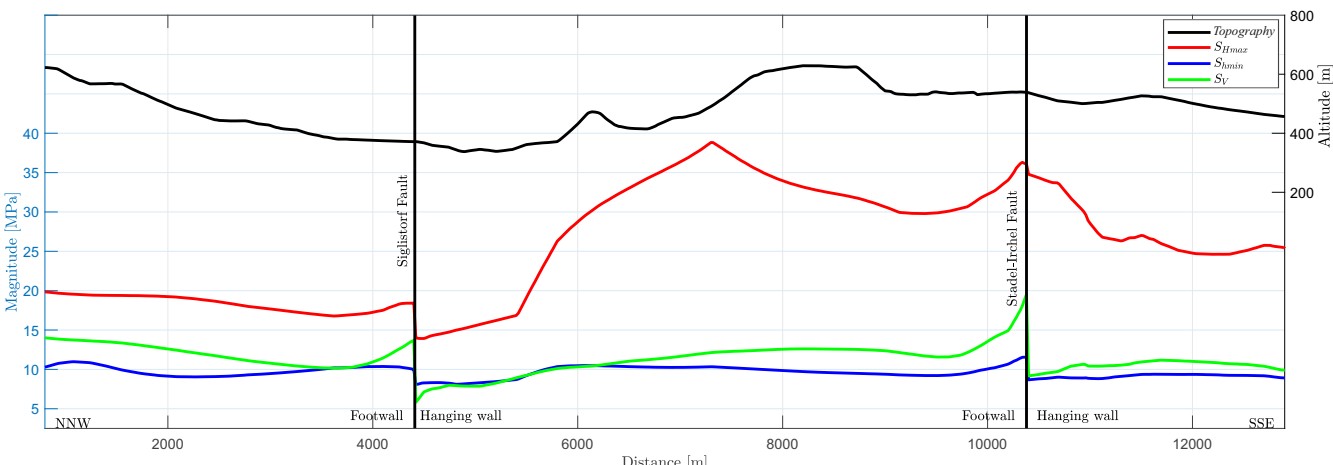

**Figure 2.** Plot of stress components along a NNW-SSE profile at approximately 400 m depth (sea level) within the Nördlich Lägern model (Hergert et al., 2015). The largest and smallest horizontal principal stress ($S_{Hmax}$ and $S_{hmin}$), the vertical stress ($S_V$) are shown. Additional shown is the topography from the model. The location of the implemented Siglistdorf- and Stadel-Irchel fault are indicated by black vertical lines. Stress magnitudes changes are significant next to the faults, but stresses are also variable due to a variable topography, rock stiffness or other factors. The significant variation in $S_{Hmax}$ is attributed to material changes, as the stratigraphic boundaries dip slightly towards the south.

The only method to test this are generic models, using geomechanical numerical methods. There are several technical methods available to represent faults or fault zones numerically; for method overview see Henk (2020). Where using the continuum method, a fault is represented by selected elements with different behaviour, e.g., a lower Young's modulus (e.g., Cappa and Rutqvist, 2011), a plastic behaviour (e.g., Mohr-Cloulomb) or viscous behaviour. In contrast to that, using the discontinue method, the fault is represented by contact elements (e.g., Buchmann and Connolly, 2007; Hergert et al., 2015) which allow offset along these structures (Fig. 1, Tab. 1). The Finite Element Method (FEM) is often used for such studies. Another discontinuous method, where the geometry is divided into several individual elements (circles or spheres, etc.) is the Discrete Element Method (DEM, e.g., Cundall and Hart, 1992; Yoon et al., 2014), which will not be used here. Physical models, using a photo-elastic material (e.g. de Joussineau et al., 2003), are also an option.

The impact of faults has also been investigated by several authors using forward models. These studies (e.g. Tab. 1) either focus on how to technically implement faults into geomechanical-numerical models (Prévost and Sukumar, 2016; Treffeisen and Henk, 2020b) or on specific geological settings (Chéry et al., 2004; Fitzenz and Miller, 2001; Hergert and Heidbach, 2011;

Meier et al., 2017; Yoon et al., 2017). As an example, Fig. 2 plots stress components along a horizontal line at sea level within a model from northern Switzerland (Hergert et al., 2015). The magnitudes of the stress tensor vary significant close to the faults.

However, resulting stress changes are affected by other factors too, such as topography or variable material properties.

Previous studies show that faults have certainly an impact, but a systematic approach is still missing. They do not provide a quantification, which component of the stress tensor is affected by the stress changes near the fault. In this paper we investigate systematically the change of individual stress tensor components with distance to the fault. In particular we determine the changes of the maximum and minimum horizontal stress ($S_{Hmax}$, $S_{hmin}$), the vertical stress ($S_V$) and the von Mises stress

as well as the orientation of the stress tensor by means of the $S_{Hmax}$ azimuth in different settings regarding fault- and rock properties, stress regime and fault structure. Again, our focus is the far-field perspective, i.e. at distances far beyond 100 m from the fault core (Fig. 1). Thus, this work does not aim to answer the question to what extend the stress tensor components are affected in the near field.

## 2 Model set-up

### 2.1 Model concept

We set up generic 3-D models with model dimensions, rock properties and an initial stress state that are like the one from a 3-D geomechanical-numerical model of a potential siting area for a high-level radioactive waste disposal site in Northern Switzerland, presented by Hergert et al. (2015). For implementation in the model, faults are represented by contact elements, which allow an offset, or 3-D elements which are elastically or plastically weaker than the surrounding rocks (Fig. 1).

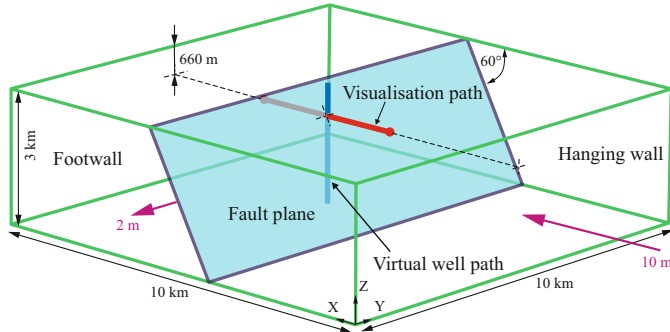

**Figure 3.** The model extent ($10 \times 10 \times 3$ km$^3$, in green) with the fault (blue plane), inclined by $60°$ (dip angle), in red the visualisation path at a depth of 660 m along which the stress magnitudes are presented for the majority of the figures. The blue vertical line indicates the location of a virtual vertical borehole (Fig. 4). The displacement boundary conditions in purple are 10 m shortening ($\epsilon = -1 * 10^{-3}$) in X-direction (perpendicular to the strike direction of the fault), which governs the $S_{Hmax}$ magnitude, and 2 m of dilation ($\epsilon = 2 * 10^{-4}$) in Y-direction (parallel to the strike direction of the fault), which drives the $S_{hmin}$ magnitude.

 **2.2 Partial differential equation and solution scheme**

The two key components of a static stress state are a result from volume forces due to gravity and surface forces from plate tectonics. Neglecting acceleration, the resulting partial differential equation is the equilibrium of forces. For the upper crust assuming linear isotropic elasticity is a good approximation to describe the stress-strain relation (e.g., Tesauro et al., 2012). Thus, for simplicity the three key model parameters in our study are density ($\rho$), the Young's modulus (E) and the Poisson's ratio ($\nu$). Additionally, the Mohr-Coulomb criteria, using the friction ($\mu$) and the cohesion (C), will be used for some models. As we introduce a fault in our model with different techniques, we solve the problem numerically using the FEM.

**2.3 Geometry and material properties**

The reference model has an extent of 10 km in each horizontal and 3 km in vertical direction (Fig. 3). The model is intersected in its entirety in the centre by a 60° inclined fault, represented by cohesionless contact elements with a friction coefficient of $\mu = 0.4$ (friction angle $\phi =$ 21.8°). The main shortening direction is perpendicular to the strike of the fault. Homogeneous linear elastic and isotropic material properties are assigned to the reference model, having a Young's modulus of $E = 15$ GPa, a Poisson's ratio of $\nu = 0.27$ and a density of $\rho = 2550$ kg m$^{-3}$. The FE-mesh for the reference model has a resolution of 50 m in the X- and Z-direction, and 500 m in the Y-direction. The mesh was created with HyperMesh 2017 and 2019 respectively; the used solver is Abaqus 6.14.1.

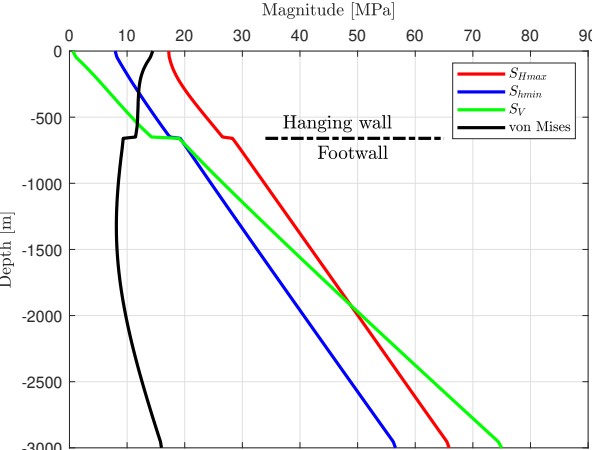

**Figure 4.** Virtual vertical well path in the centre of the reference model. Shown are the resulting stress components, which are $S_{Hmax}$, $S_{hmin}$, $S_V$ and the von Mises stress. The fault with a friction coefficient of $\mu = 0.4$ is traversed at a depth of -660 m. It is visible that when crossing the fault from the hanging wall to the footwall block, there is a sudden increase of $S_V$, for $S_{Hmax}$ and $S_{hmin}$ a little less.

 **2.4 Model scenarios**

The scope of the study is to investigate factors that affect the stress state in the broader vicinity of faults. These include the element resolution (pre-tests), the representation of the fault by contact elements with a variable friction coefficient, representation of the fault by elastic weaker elements, or by elements with elasto-plastic rheology, the inclination of the fault, the strike direction relative to the shortening direction, the variation of the rock stiffness (Young's modulus) and the size of the fault and model itself. In order to allow a good readability of the study, specific variations of the model are always briefly explained before presenting the modelling results.

**2.5 Initial stress state and boundary conditions**

We implement an initial stress state of the model that is in equilibrium with the gravitational forces without resulting in any significant displacement along the fault and the model geometry. We follow the technical procedure as explained in Hergert et al. (2015). In a second step we apply along the model lateral boundaries displacement boundary conditions that result in tectonic stresses throughout the model volume. The main shortening of the reference model is perpendicular to the fault (X-direction) in the order of -10 m ($\epsilon = 1 * 10^{-3}$), which then corresponds to the $S_{Hmax}$ orientation. Parallel to the fault strike (Y-direction), the model undergoes a slight dilation of 2 m ($\epsilon = 2 * 10^{-4}$), which is then the orientation of $S_{hmin}$ (Fig. 3). The stress magnitudes resulting from the boundary conditions are shown in Fig. 4 along a vertical synthetic well path in the centre of the model. This stress state is in general agreement with stress magnitude data that were derived from a measurement campaign in Northern Switzerland using >150 micro-hydraulic fracturing and sleeve reopening tests (Desroches et al., 2021).

**2.6 Stress definition and visualisation**

The 3-D stress state of the Earth's crust is described by a second rank tensor ($\boldsymbol{\sigma}$, Jaeger et al., 2011) with nine components, but due to its symmetry only six components are independent from each other. As common in geoscience, compressive stress magnitudes are positive and tensile stresses are negative. The stress state can also be described with the magnitudes and orientations of the three principal stresses. These principal stresses are named from the largest to the smallest as $\sigma_1 > \sigma_2 > \sigma_3$.

Furthermore, in our model the vertical stress ($S_V$) is a principal stress (Eq. 1). As a result, the two other principal stresses are in the horizontal plane and are labelled as the minimum and maximum horizontal stresses ($S_{hmin}$ and $S_{Hmax}$).

$$\boldsymbol{S}_V = \int_0^z \rho g z, \tag{1}$$

The relative ratio of these three principal stresses defines the stress regime (Anderson, 1905, 1951):

| Normal faulting stress regime | NF | $S_V > S_{Hmax} > S_{hmin}$ |
| Strike-slip stress regime | SS | $S_{Hmax} > S_V > S_{hmin}$ |
| Thrust faulting stress regime | TF | $S_{Hmax} > S_{hmin} > S_V$ |

Additionally, we use the differential stress ($\sigma_D$) and it's 3-D-equivalent, the von Mises stress ($\sigma_{vM}$; Mises, 1913) to visualise the stress state (Eqs. 2 and 3).

$$\sigma_D = \sigma_1 - \sigma_3 \tag{2}$$

$$\sigma_{vM} = \sqrt{\frac{1}{2}(\sigma_1 - \sigma_2)^2 + (\sigma_2 - \sigma_3)^2 + (\sigma_3 - \sigma_1)^2} \tag{3}$$

The model results are presented here in the same way whenever possible. Both, the stresses components $S_{Hmax}$, $S_{hmin}$, $S_V$ and the von Mises stress are used to visualise the influence of a fault on the stress state. The results of the models are plotted along a horizontal path at a depth of -660 m (Figs. 3 and 5). This path is always parallel to the main shortening direction (X), expect for the models witch a variable fault strike. The visualisation extends from the footwall block at $-3000$ m through the fault at 0 m to $+3000$ m in the hanging wall block.

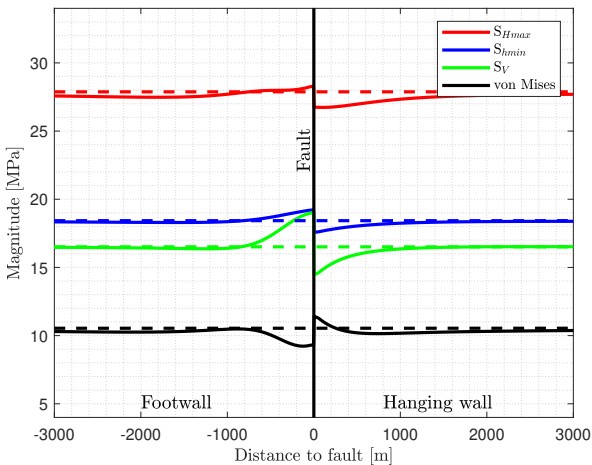

**Figure 5.** Stress magnitude visualisation of the reference model from $-3000$ m (footwall – left) to $+3000$ m (hanging wall – right) for a constant depth of -660 m. Used are linear elastic material properties and a friction coefficient of $\mu = 0.4$ for the fault at 0 m, represented by the black vertical line. The dashed lines represent in comparison results of an similar model without a fault.

**2.7    Pre-test: Mesh resolution**

The impact of the mesh resolution and sufficiency is investigated by varying the mesh size, using elastic material properties only, like the reference model. An mesh resolution of 1000, 500, 250 and 100 m in all directions is tested; a finer resolution has been used with an element size of 50 m in the main shortening and depth direction (X and Z), for which the resolution parallel to the fault (Y) is 500 m.

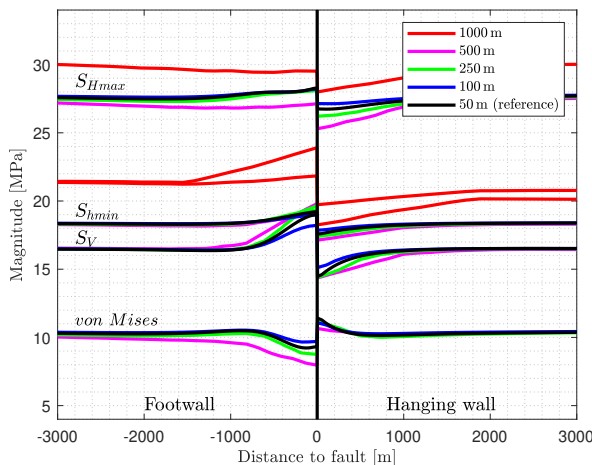

**Figure 6.** The impact of the mesh resolution is compared. The coefficient of friction of the fault is $\mu = 0.4$ for all models. The coarse resolution models, with 1000 m (red) as well as 500 m (magenta) show significant deviations from the reference model with a resolution of 50 m (black), while the models with a resolution of 100 m and 250 m (green and blue) show only slight deviations close to the fault.

The model with the coarsest resolution (1000 m) provides stress magnitudes that deviate significantly from the other models (red line in Fig. 6). Even for the model with a resolution of 500 m (magenta line in Fig. 6), the deviation from the higher-resolution models, at a distance greater than 1000 m is clearly visible. All finer-resolution models ($\leq$250 m), have only small differences close to the fault ($<$1000 m; Fig. 6). This shows that all models with a resolution of 250 m and finer have a sufficient mesh resolution. A finer mesh is only useful if the stress changes close to the fault is of interest, which is not the case in this study.

## 3 Results

### 3.1 Reference model

Within the reference model, the fault is represented by a contact surface ($\mu = 0.4$, $C = 0$). As a result, the components of the reduced stress tensor increase in the footwall close to the fault and decrease in the hanging wall (Fig. 5). $S_V$ and $S_{hmin}$ rise to a similar level (+3 MPa and +1 MPa) within the footwall block near the fault. An opposite behaviour is observed for the von Mises stress. $S_{Hmax}$, however, increases only slightly close to the fault ($<$1 MPa); which is the reason for the decrease of the von Mises stress near the fault. Corresponding to these changes, the stress magnitudes decrease next to the fault within the hanging wall block, the largest amount is for $S_V$, resulting in a slight increase of the von Mises stress. Significant stress changes of more than 1 MPa occur within a distance of 1000 m from the fault. The $S_{Hmax}$ orientation is not affected by the fault.

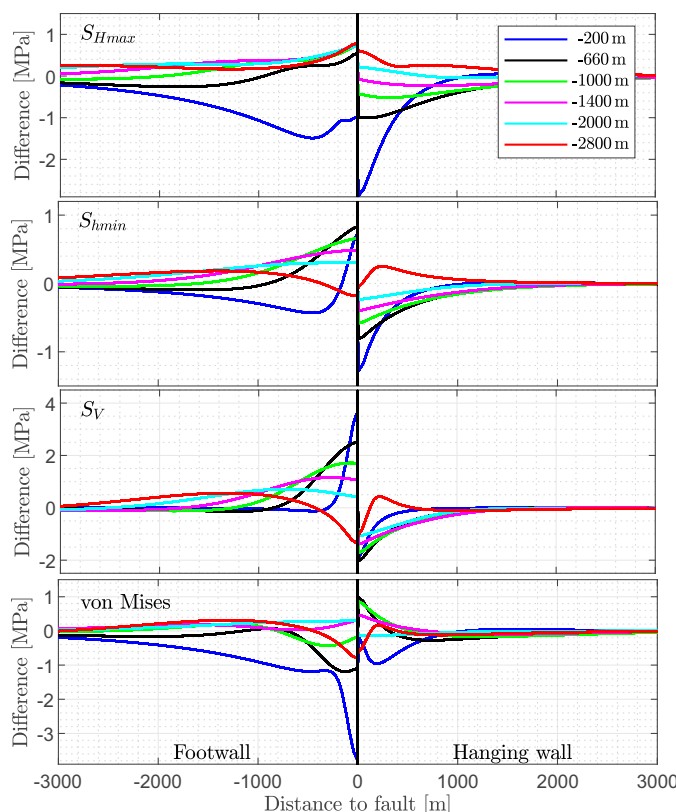

**Figure 7.** Variation of the stress components ($S_{Hmax}$, $S_{hmin}$, $S_V$ and von Mises stress) at different depth levels are shown with respect to the distance of the fault. Stress magnitude changes are visualised along a vertical line at depths of $-200$, $-660$ (reference depth, used by the other figures), $-1000$, $-1400$, $-1400$, $-2000$ and $-2800$ m.

The results of all other models presented subsequently are displayed on a horizontal path at the same depth. For the reference model, the stress variation around the fault for different depth ranges are also shown in Fig. 7. It remains unchanged that stress variations $>1$ MPa are limited to a distance of about 1000 m from the fault. Relatively large variations can be seen at shallow depths (blue, $-200$ m) in contrast to greater depths (red, $-2800$ m). The general patterns of stress variation are similar, except for the vertical stress component. $S_V$ is smaller in the footwall block close to the fault, and larger in the hanging wall block at a depth of $-2800$ m, in contrast to observation at shallower depth ($<2000$ m). The reason is that $S_V$ becomes $\sigma_1$ (normal faulting regime) for a depth greater than 2000 m, while at shallower depths a transition from a thrust faulting to a strike-slip regime occurs (Fig. 4).

## 3.2 Friction coefficient

In geomechanics and seismology faults are usually parameterised using the friction coefficient and the cohesion (e.g. Morris et al., 1996; Di Toro et al., 2011; Röckel et al., 2022). Commonly, a friction coefficient between 0.6 and 0.85 is assumed

(Byerlee, 1978) but examples exist of significantly smaller friction coefficients (Di Toro et al., 2011). However, to investigate the influence of the frictional properties of a fault based on the reference model, the friction coefficient is varied from very low ($\mu = 0.1$) to very large ($\mu > 1$).

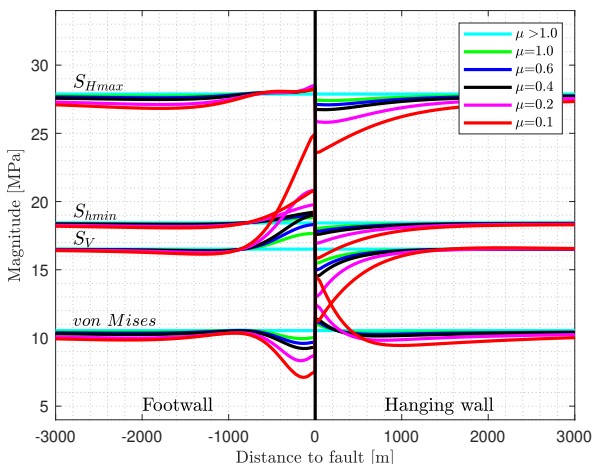

**Figure 8.** Impact of a variable friction coefficient on the stress state. Plotted are the $S_{Hmax}$, $S_{hmin}$ and $S_V$ as well as the von Mises stress. The graph with the friction angle of $\mu = 0.4$ is the reference model (Fig. 5). Large stress variations near the fault are a result from low friction.

Using a very large friction coefficient ($\mu > 1$), there is no visible influence by the fault on the stress magnitudes (Fig. 8), the
stress magnitudes are identical to a continuous mesh without a contact surface (dashed line in Fig. 5). In contrast, for a low friction case ($\mu = 0.1$), stress variation is significant near the fault. The general pattern is similar as for the reference model, but the increase (footwall: $+8$ MPa) and decrease (hanging wall: $-5$ MPa) of $S_V$ is much larger. Similar, but not that large $S_{hmin}$ changes of $+2$ MPa are observed for for the footwall und $-2$ MPa for the hanging wall block. The drop of $S_{Hmax}$ in the hanging wall block is significant ($-4$ MPa), whereas the increase in the footwall block next to the fault is negligible. However, a $S_{Hmax}$
decrease of about $-1$ MPa is visible in both, the footwall- and hanging wall block, even between 1000 to 3000 m away from the fault. This is a result of stress dissipation due to larger fault offset in the case of low friction. Variation of the von Mises stress is mainly driven by the variation of $S_V$. It is mostly $\sigma_3$, trimmed by the fact, that $S_V$ becomes significant larger then $S_{hmin}$ in the footwall block about 500 m next to the fault for the models with low friction contact definition.

Overall comparison of the models with a different friction in Fig. 8 show, that the stress perturbations gradually decrease
with an increase of the friction coefficient. A stress variation of $> 1$ MPa is limited to a distance of $\approx 1$ km, except for $S_{Hmax}$ in the hanging wall block. None of the variation result in a visible change of the $S_{Hmax}$ orientation, it is always parallel to the maximum displacement (X-direction).

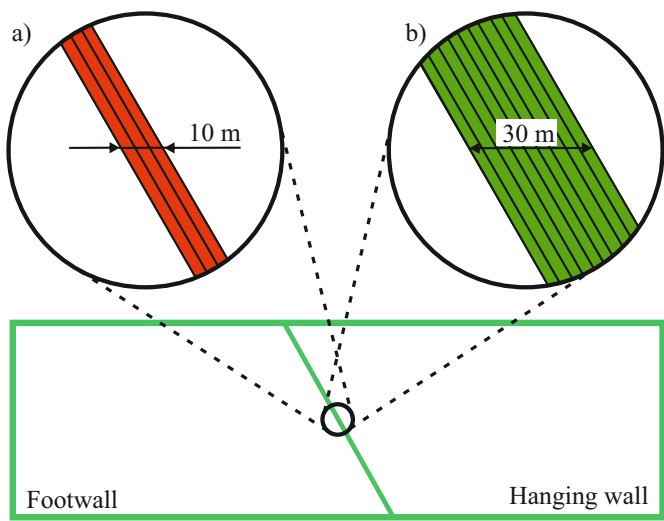

**Figure 9.** Sketch visualising the representation of the fault zone by elastically weak 3-D-elements with a thickness of a) 10 m made from three elements or b) 30 m made of nine elements. The elements outside this fault zone are not visualised.

### 3.3   3-D-fault representation by elastic weak elements

The representation of a fault by a 2-D plane is not realistic for the immediate vicinity of the fault where a zone of damaged rock is expected. A more realistic approach seems to be the representation by a layer of elements with an elastic rheology of reduced stiffness (Fig. 9). This simulates the damage zone around the fault core (e.g., Faulkner et al., 2006).

Herein, the fault zone has a width of 10 m represented by three elements normal to the fault (Fig. 9 a). A Young's modulus of $E =$ 5, 1 and 0.25 GPa, is tested while the stiffer surrounding has $E =$ 15 GPa. The element resolution outside the fault area is 50 m in X- and Z-direction and 500 m in the Y-direction.

The stress magnitudes along the profile (Fig. 10) do not show a significant stress variability in the vicinity of the fault resulting from three the less stiff elements. Stress changes are restricted to a very narrow domain, which are not visible; they are visually hidden behind the fault line. $S_{Hmax}$ decreases depending on the decreasing stiffness. For the model with $E =$ 250 MPa fault representation, $S_{Hmax}$ is always around 1 MPa lower because of stress dissipation by the low stiff fault domain. Therefore, the von Mises stress drops by the same amount. Stress dissipation also effects $S_{hmin}$, but with a much lower amount; for $S_V$ no effect is visible.

Another model version has a thicker fault of 30 m, represented by nine elements normal to the fault (Fig. 9 b). Like the 10 m models, $S_{Hmax}$ drops especially for the model with the least stiff fault domain ($E =$ 250 MPa) by around -3 MPa (Fig. 11), again an effect of the stress dissipation. $S_{hmin}$ decreases by almost 1 MPa, whereas $S_V$ is stable. Near the fault, $S_{Hmax}$, $S_{hmin}$ and $S_V$ decrease significant, limited to a region, narrow to the fault (<100 m). The von Mises stress variation is mainly driven by the reduction of $S_{Hmax}$ because of the less stiff fault parts. There is no change of the $S_{Hmax}$ orientation to observe and it remains parallel to the X-direction.

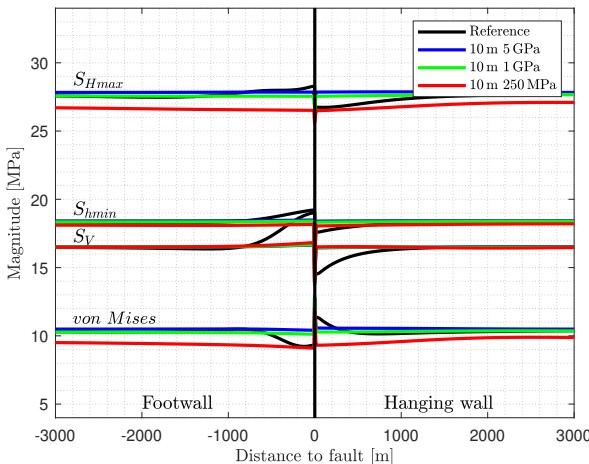

**Figure 10.** Fault representation by a 10 m thin layer of three weak elements. The fault elements have a lower Young's modulus ($E$ =5, 1 and 0.25 GPa) in contrast to the area outside this region ($E$ =15 GPa). Shown in black is the reference model, and vertically the implemented fault zone. Stress changes are narrowly limited to the area of the fault so that they are hidden by the visualisation of the fault zone.

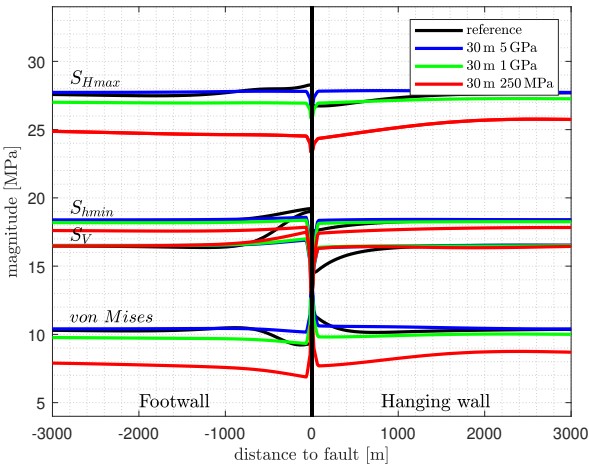

**Figure 11.** Fault representation by 30 m (nine elements) of elastic weak elements having a lower Young's modulus ($E$ =5, 1 and 0.250 GPa), compared to the area outside this region with $E$ =15 GPa. In colours are the model results with the less stiff 3-D-fault representation. Shown in black is the reference model using contact surfaces, and vertically the implemented fault zone at 0 m.

### 3.4 3-D-fault representation by elements with elasto-plastic rheology

As purely elastic elements do not allow failure, they cannot dissipate stresses such as a contact surface is able to do. To accommodate both, the ability to dissipate stresses and the representation of a damage zone, elements with elasto-plastic

rheology within the fault zone are now used. Out of a continuous mesh, elements close to the fault location were selected
in a staircase-like manner, which have a specific plastic yield criterion. These fault elements have laterally a range of one
(Fig. 12 a) to eight elements (Fig. 12 b). These elements have a friction angle of $\phi = 30°$ (friction coefficient $\mu = 0.58$) and a
low cohesion of $C = 0.1$ kPa. The used dilation angle is $\psi = 25°$. In contrast to that, the non-fault elements have a much larger
cohesion ($C = 500$ kPa), but the same friction- and dilation angle. The element resolution in the vicinity of the fault is 100 m
in X- and Z-direction, and 500 m in the Y-direction. The elastic material properties are the same as used by the reference model
($E = 15$ GPa, $\nu = 0.27$ and $\rho = 2550$ kg m$^{-3}$).

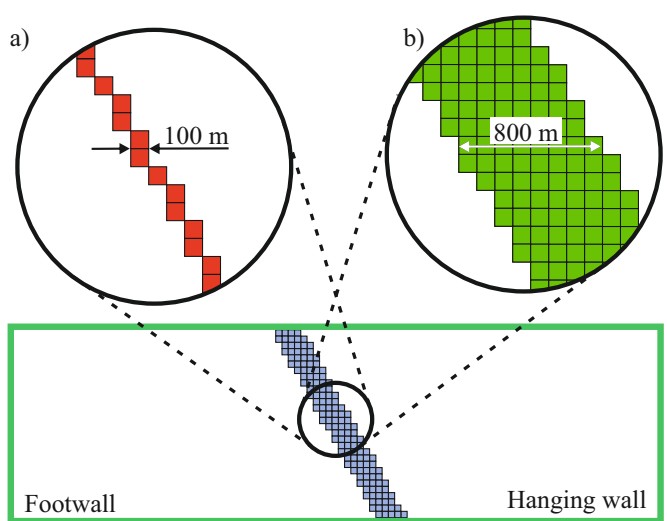

**Figure 12.** Sketch showing the fault representation by selected elements out of the mesh, which plastify as a result of friction and a low
cohesion of $C = 0.1$ kPa. Elements outside this region (white area, mesh not shown) have a cohesion of $C = 500$ kPa. A friction angle
$\phi = 30°$ (friction coefficient $\mu = 0.58$) is used for all elements for the first test. Different number of lateral elements, representing the fault
are tested, ranging from one (a) to eight (b) lateral elements. As the element size is 100 m near the fault, the total width of the stair-step like
fault ranges from 100 to 800 m.

The representation by means of staircase-like elements with elasto-plastic properties (Fig. 13) shows that the impact on the
stress components is nearly independent from the amount of laterally used elements that allow plastification. $S_{Hmax}$, $S_{hmin}$,
$S_V$ slightly increase in the footwall block near the fault domain and slightly decrease in the hanging wall block, again near
the fault domain. The overall variation of $S_{Hmax}$, $S_{hmin}$, $S_V$ and the von Mises is $<1$ MPa. Stress magnitudes do not show any
discontinuous behaviour at the fault zone, as the reference model do. Stress variations are restricted to a zone of about 1000 m
next to the fault domain. Again, the $S_{Hmax}$ orientation is not disturbed as a result of the fault.

The model having laterally four weak elements is used again to investigate the impact of the friction. Friction angles of
$\phi = 30, 25, 20$ and $15°$ are applied. The $\phi = 30°$ model has already been used for the variation of the number of lateral elements
(Fig. 13: 4 Elements). Modelling results in Fig. 14 show, that a decreasing friction angle increases the stress variation near the
fault. $S_{Hmax}$, $S_{hmin}$, $S_V$ increase in the footwall block near the fault, while a slight decrease can be seen in the hanging wall

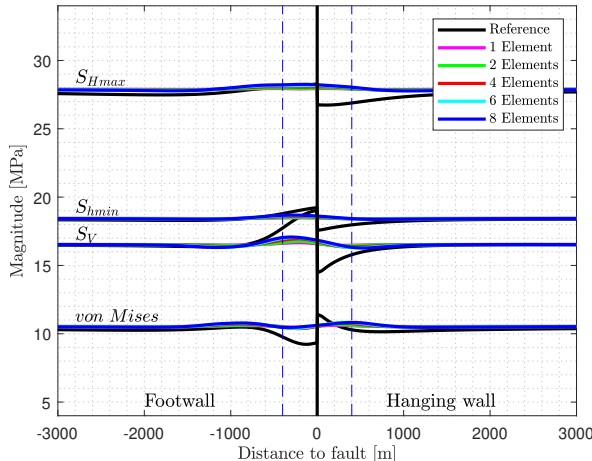

**Figure 13.** Fault representation with staircase-like elements with elasto-plastic rheology (Fig. 12) that are allowed to deform non-elastically. Shown in black is the reference model with the implemented fault, in colours are the models with a continuous mesh with the one (magenta) to eight lateral elements (dark blue). These elements have a low cohesion of $C = 0.1$ kPa and a friction angle of $\phi = 30°$ (friction coefficient $\mu = 0.58$). The maximum width of eight elements is visualised by the dashed blue lines.

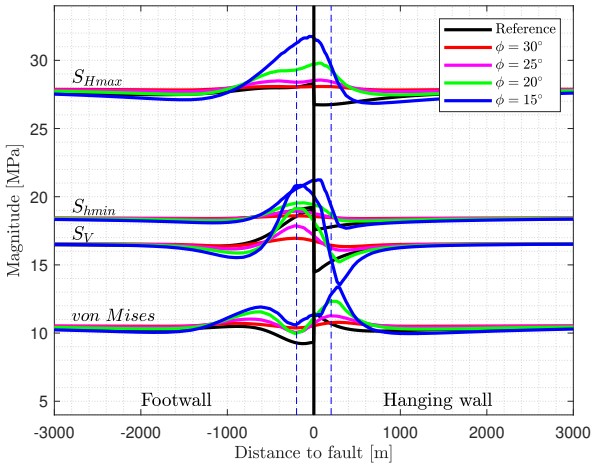

**Figure 14.** Fault representation with four staircase-like elements with elasto-plastic rheology (Fig. 12) that are allowed to deform non-elastically. Shown in black is the reference model and the fault centre (vertical), in colours are the models with a friction angle of $\phi = 30$, 25, 20 and 15°. The $\phi = 30°$ model is the same as the four element model in Figure 13. The width of four elements is visualised by the blue dashed lines.

block. However, swing-in effects can be observed on both sides of the fault. Largest magnitude changes are about +4.5 MPa

for $S_V$, +5 MPa for $S_{Hmax}$ and +2.5 MPa for $S_{hmin}$. In a distance of >1400 m, to the fault centre, the variation of the stresses is <1 MPa. The $S_{Hmax}$ orientation remains unaffected.

## 3.5 Variation of the fault dip angle

To study the impact of the fault dip angle, several models with different fault inclination are prepared. These models have an dip angle of 30°, 40°, 50°, 70° and 80°, in contrast to the reference model (60°, Fig. 3 and 5). Elastic material properties are the same as used in the reference model: $E = 15$ GPa, $\nu = 0.27$, $\rho = 2550$ kg m$^{-3}$ and fault representation by contact elements: $\mu = 0.4$ and $C = 0$.

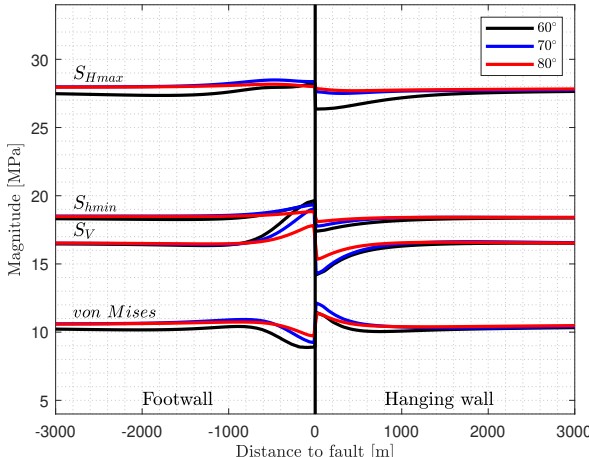

**Figure 15.** Influence of the dip angle of the fault on the stress components $S_V$, $S_{Hmax}$, $S_{hmin}$ and the von Mises stress. Shown are the models with a fault dip angle of 60° (reference model), 70° and 80°. By increasing the dip angle, the magnitude of stress perturbation decreases.

In Fig. 15, it can be seen that the stress perturbation pattern is similar compared to the reference model. With increasing dip angle from 60° (reference model) to 70° and 80°, the stress perturbation slightly decreases. The reduction is most significantly visible for the $S_V$ magnitude in the footwall block next to the fault. Stress magnitudes in a distance from the fault increase slightly for the large dip angle models, as the stress dissipation by the fault decreases.

A decrease in dip angle of the fault results in a significantly more pronounced increase of the stress perturbation near the fault (Fig. 16). This results in an increase of $S_{Hmax}$ by >4 MPa in the footwall block, and a decrease of about −4 MPa in the hanging wall block, using an fault inclination of 30°. An increase of the $S_V$ and $S_{hmin}$ magnitudes in the footwall block and a decrease in the hanging wall block is clearly visible. The influence of the fault on the $S_{Hmax}$ magnitude on both, the footwall and hanging wall block is in a distance to the fault of about 1500 m and 2000 m. However, the large distance is an effect of the small fault dip, the real distance is half of that values for the 30° model. There is no perturbation of the $S_{Hmax}$ orientation.

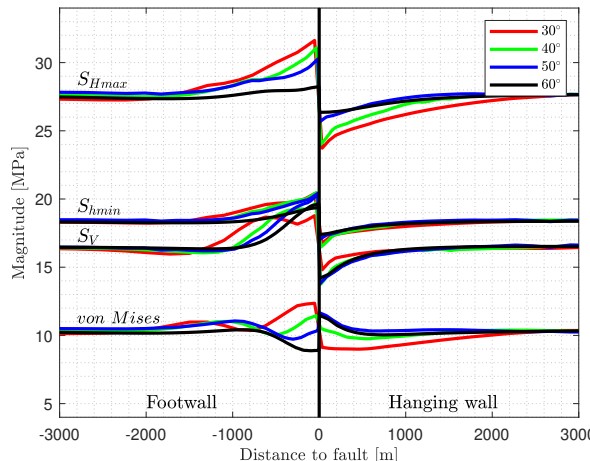

**Figure 16.** Influence of the fault dip angle on the stress components. A range of fault dip angles are presented, including 60 (reference model), 50, 40 and 30°. By reducing the dip angle, the stress magnitude changes and the distance of the lateral stress perturbation increases. The most pronounced stress perturbation is seen for shallow dipping faults (30° in red).

### 235   3.6   Variation fault strike angle

In addition to the influence of the dip, the influence of the $S_{Hmax}$ orientation with respect to the fault strike is investigated. Thus, an strike angle of 90° as the reference model is compared with other models where the fault is striking with an angle of 75°, 60°, 45°, 30° and 15°. To geometrically allow such strike angles, the models are extended in the X-direction from 10 km to 20, 30 and 50 km for the models with an fault strike of 45°, 30° and 15° respectively. The resulting boundary conditions are
adjusted, to ensure comparability.

Results of the strike angle variation (Fig. 17) are shown perpendicular to the strike direction of the fault. The impact of the fault strike variation on the $S_{Hmax}$ and $S_{hmin}$ magnitude is minimal. Clear deviations are only observed for $S_V$ in the footwall block, where $S_V$ is smaller compared to the reference model. As a result, the von Mises stress is also less variable in the footwall block, next to the fault The variation of the $S_{Hmax}$ orientation varies with distance to the fault, but does not exceed 1.5°, which
is significantly smaller than the uncertainties of orientation data records (Heidbach et al., 2018). Therefore, no visualisation of that is shown.

Since the models with the fault strike variation and the friction coefficient of 0.4 only cause small $S_{Hmax}$ rotations, the influence of a lower friction ($\mu = 0.1$) is also investigated. The plot of the stress magnitudes (Fig. 18) shows a visible variation of the magnitudes for the different orientations of the fault. The general pattern is similar to the reference model. For $S_{Hmax}$,
significant variations in stress magnitude are observed between the models due to stress dissipation resulting from low friction at the fault. The largest magnitudes are for the reference model (90°) as well as the 15° model. In contrast, the 45°, 30° and 60°

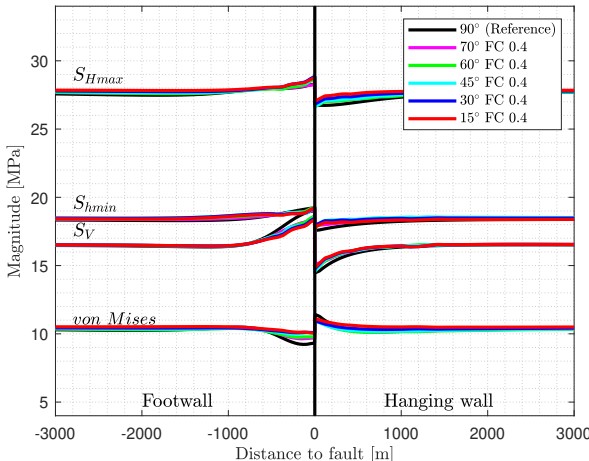

**Figure 17.** Stress components are shown for the models with a variation of the strike angle, relative to the orientation of the maximum shortening using a friction coefficient of $\mu = 0.4$. In contrast to the reference model with a fault strike angle of $90°$, the varied models have a strike angle of $75°$, $60°$, $45°$, $30°$ and $15°$. The stress components are plotted perpendicular to the strike of the fault.

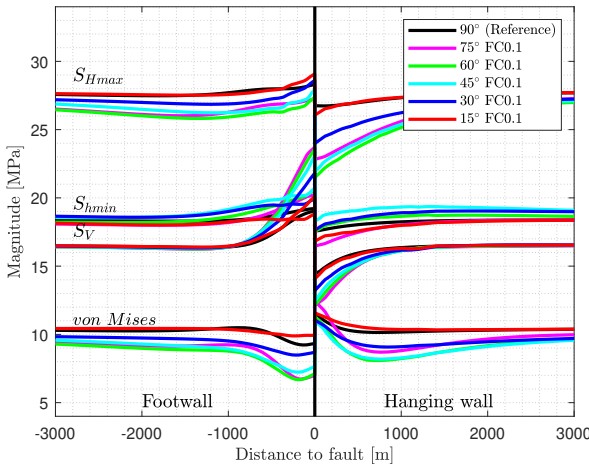

**Figure 18.** Variation of the strike angle, with 90 (reference), $75°$, $60°$, $45°$, $30°$ and $15°$ relative to the orientation of the direction of maximum shortening. Used is a friction coefficient of $\mu = 0.1$ in contrast to the similar models with a friction coefficient of $\mu = 0.4$ (Fig. 17). The stress components are plotted perpendicular to the strike of the fault.

models have the largest $S_{hmin}$ magnitudes. As a result of the largest variation of the $S_{Hmax}$ magnitudes, the lowest von Mises stresses are observed for 45, 30 and $60°$ models.

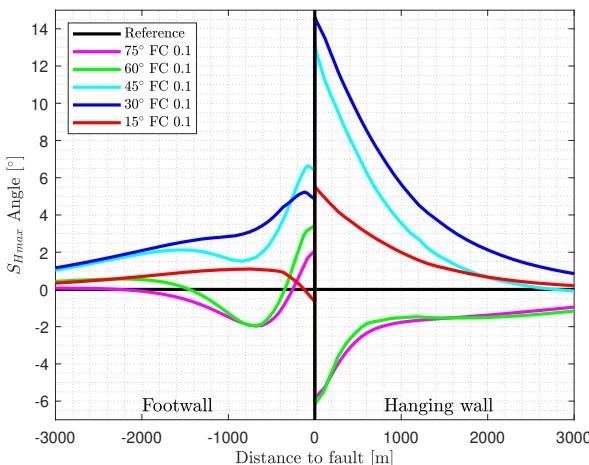

**Figure 19.** Variation of the strike angle ($75°$, $60°$, $45°$, $30°$ and $15°$), relative to the orientation of the maximum shortening direction using a friction coefficient of 0.1. Shown are the variation of the $S_{Hmax}$ orientation, compared to the reference model with a fault strike angle of $90°$ and an constant $S_{Hmax}$ orientation of $0°$. The angular variation is plotted perpendicular to the strike of the fault.

For the first time in the model series, a significant variation in the orientation of $S_{Hmax}$ is clearly visible with a fault strike variation using a friction coefficient of $\mu = 0.1$ (Fig. 19). The deviation of the orientation reaches up to about $14°$ in the hanging wall block for the model with an fault strike of $30°$, closely followed by the $45°$ model. The $S_{Hmax}$ rotation for the $30°$, $45°$ and $15°$ models is clockwise, parallel to the strike of the fault, while in the models with a strike of the fault of $60°$ as well as $75°$, $S_{Hmax}$ orientation is counterclockwise, i.e. tends to be perpendicular to the orientation of the fault. In the footwall block, the rotation of $S_{Hmax}$ is also visible, but less than in the hanging wall block, with maximum of about $6°$.

## 3.7 Young's Modulus

Since the elastic material properties have a significant influence on the deformation on the rock of both sides of the fault, the Young's modulus of the host rock is varied. In addition to the Young's modulus of the reference model ($E = 15$ GPa), stiffnesses of 5, 20, 30, 40, 60, 80 and 100 GPa are tested. In order to keep the model comparable, the boundary conditions are adapted (Tab. 2), so that the far-field stress magnitudes of the different models were equal.

The variation of the Young's modulus has limited effect on $S_{Hmax}$ in the footwall block (Fig. 20), where in the hanging wall block $S_{Hmax}$ decreases by up to $-4$ MPa with increasing Young's modulus next to the fault. $S_{hmin}$ increases slightly with the Young's modulus in the footwall block, and decreases in the same way in the hanging wall block slightly by up to $-2$ MPa. The $S_V$ magnitude shows the same pattern, but the stress deviation is much larger near the fault, up to $+7$ MPa in the footwall and $-4.5$ MPa in the hanging wall block. The von Mises stresses decrease with increasing Young's modulus in the footwall block next to the fault and increases in the hanging wall block next to the fault.

**Table 2.** Boundary conditions are chosen depending on the Young's modulus to generate equal far-field stress magnitudes for the different models. The boundary conditions for 15 GPa are the reference model settings.

| Young's Modulus [GPa] | X-shortening [m] | Y-dilation [m] |
|---|---|---|
| 5 | 30.000 | 6.000 |
| 15 | 10.000 | 2.000 |
| 20 | 7.500 | 1.500 |
| 30 | 5.000 | 1.000 |
| 40 | 3.750 | 0.750 |
| 60 | 2.500 | 0.500 |
| 80 | 1.875 | 0.375 |
| 100 | 1.500 | 0.300 |

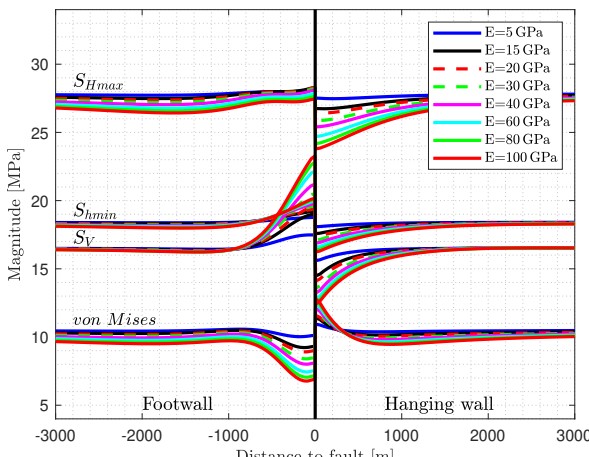

**Figure 20.** The influence of the Young's modulus on the stress perturbation is investigated. The models have a Young's modulus of $E = 5$, 15 (reference model), 20, 30, 40, 60, 80 and 100 GPa.

In general, the stress perturbation increases due to a larger Young's modulus, as stress dissipates on the fault. The lateral influence of the fault on the stress components, producing a stress variation of more than 1 MPa, is limited to a range from $-1000$ m to $+1000$ m next to the fault. Again, the $S_{Hmax}$ orientation is always parallel to the direction of principal shortening.

## 3.8 Model size

It is obvious that the influence of the fault on the stress state also depends on the size of the fault surface or on the overall size of the model. For this purpose, the size of the active fault surface using the reference model geometry is reduced to

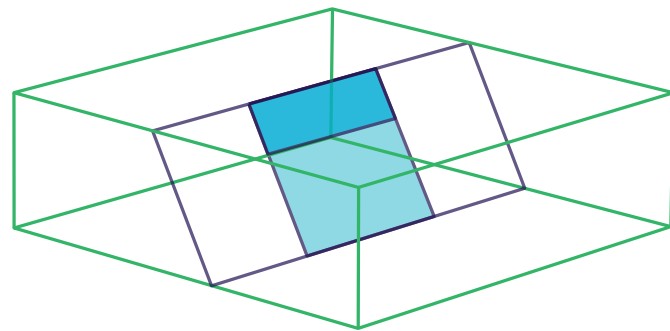

**Figure 21.** Model sketch with a reduced fault surface area of $4 \times 3 \, \text{km}^2$ (light and dark blue areas together) and $4 \times 1 \, \text{km}^2$ size (dark blue area only). Everything else is just the same as shown by Fig. 3.

$4000 \times 1000 \, \text{m}^2$ and $4000 \times 3000 \, \text{m}^2$ (Fig. 21). Also, the reference model with the full fault surface is doubled and quadrupled in size. The resulting models then have dimensions of $20 \times 20 \times 6 \, \text{km}^3$ and $40 \times 40 \times 12 \, \text{km}^3$, respectively. The resulting mesh resolution is then $100 \, \text{m}$ and $200 \, \text{m}$ in the X- and Z-directions, respectively, and 1 and 2 km in the strike direction (Y) of the fault, which is parallel to $S_{hmin}$. The boundary condition were adjusted accordingly, to generate a similar stress state.

The comparison of the results in Fig. 22 shows that as the size of the fault increases, the magnitude deviation near the fault increases. Thus, in the hanging wall block $S_{Hmax}$ is reduced by almost $-3 \, \text{MPa}$, while $S_V$ in the footwall block is increases by more than $+5 \, \text{MPa}$ for the model with side length of 40 km. As a result, the von Mises stress in the footwall block decreases more significantly close to the fault. However, the increase of the fault surface area does not have a significant influence on the far-field stress pattern. Significant stress changes ($>1 \, \text{MPa}$) occur up to about 1000 m next to the fault. No rotation of the $S_{Hmax}$ orientation can be observed.

### 3.9 Strain variation

The effect of stress anisotropy is studied by defining variable lateral boundary conditions. The shortening, perpendicular to the fault strike (X-direction) is tested from 1, 2, 3, 4, 6, 8, 10 (reference model), 12, 14, 16 and 20 m ($\epsilon$ = -1*10$^{-4}$ to $-2*10^{-3}$), where the dilation to the fault (Y-direction) remains identical to the reference model of $-2 \, \text{m}$ ($\epsilon = 2*10^{-4}$). Everything else is identical to the reference model.

The different $S_{Hmax}$ magnitudes result directly from the variable shortening, applied to the model boundaries (Fig. 23). The overall pattern is like the reference model. The observed variation is low for low strain, where variation is larger for higher strain. $S_{Hmax}$ is smaller for lager strain away from the fault and increases a bit next to the fault. In the footwall block, the pattern is clear: the closer to the fault, the smaller is $S_{Hmax}$.

The variation of $S_{hmin}$ is similar to $S_{Hmax}$, variation is small for less shortening and increases by increasing shortening of the model (Fig. 23). $S_{hmin}$ increases in the footwall block next to the fault and is smaller next to the fault in the hanging wall block.

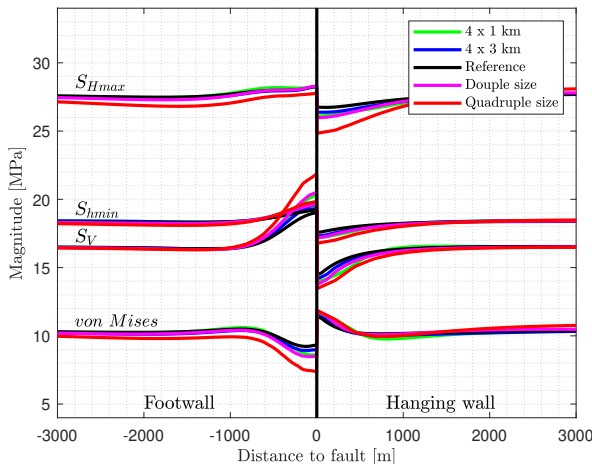

**Figure 22.** Influence of the fault size on the stress components $S_{Hmax}$, $S_{hmin}$, $S_V$ and the von Mises stress. Models with a reduced fault surface area with a size of $4{\times}3\,km^2$ and $4{\times}1\,km^2$ (Fig. 21), as well as models like the reference model with a total size of $20{\times}20{\times}6\,km^3$ (double size) and $40{\times}40{\times}12\,km^3$ (quadruple size) are shown.

Larger variation can be seen for $S_V$, with an increase in the footwall bock and a decrease in the hanging wall block. The $S_V$ magnitude variation in the footwall block increases from $+0.4\,MPa$ for 2 m of shortening to $+2.3\,MPa$ for 20 m of shortening.
Nearly the similar amount of decrease happens in the hanging wall block.

The von Mises stress variation (Fig. 23) increases with the increase of shortening compared to the reference model. For the model with little strain (<4 m) the observed variation of the von Mises stress displays another pattern. For them, the von Mises stress increases in the footwall block and decrease in the hanging wall block, next to the fault. Again, major stress variations are limited to a distance of less than 1000 m next to the fault. The $S_{Hmax}$ orientation is not affected for larger
shortening perpendicular to the fault. For the models with a shortening of 1 and 2 m in the X-direction, the stress magnitudes is horizontally isotropic ($S_{hmin} = S_{Hmax}$) and the $S_{Hmax}$ orientation is not clearly defined.

## 4    Discussion

### 4.1    Model set up and assumptions

The goal is to investigates the impact of faults on the far-field stress state (>100 m). The model design does not allow estima-
tions on the stress state or stress perturbations close to a fault (<100 m). Investigating that, a much finer mesh resolution would be needed. It is also questionable whether and which methods of fault implementation are suitable for this purpose.

Like all generic models, those ones used here are a significant simplification of rock physics, geological structures, and the fault representation itself. Except for two scenarios, only linear elastic material properties are used to represent the rock volume.

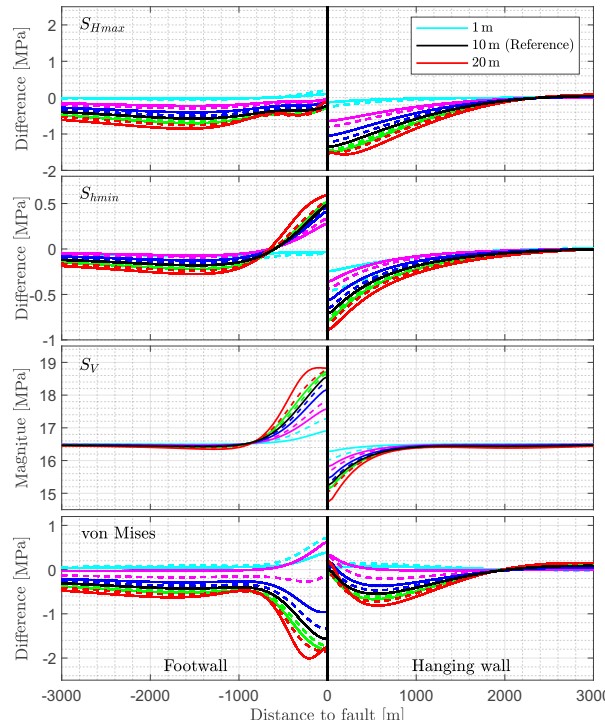

**Figure 23.** Influence of a variable strain on the stress components are shown. The models have a shortening of 1, 2, 3, 4, 6, 8, 10 (reference), 12, 14, 16 and 20 m ($\epsilon$ = -1*10$^{-4}$ to $-2*10^{-3}$) perpendicular to the strike of the fault (X-direction) and a constant dilation of 2 m ($\epsilon$ =2*10$^{-4}$) parallel to the fault (Y-direction). Avoiding an overload on the legend, only the 1, 10 and 20 m models are indicated there. As the different lateral strain along the model boundaries result in different stress magnitudes, only the relative stress changes (local stresses − far-field stress) are shown for $S_{Hmax}$, $S_{hmin}$ and the von Mises stress. The general pattern of stress variation is like the reference model, the variation is smaller for lesser strain and larger for more strain. However, relative variation of the stress components are not bigger as about 1.5 MPa for $S_{Hmax}$, $S_{hmin}$, around 2 MPa for the von Mises stress and about 2.3 MPa for $S_V$.

This neglects various rheological processes within the Earth's crust. But Hooke's law seems to be a proper approximation for
the major mechanical behaviour of rocks in the upper crust, as the elastic thickness of the crust ($T_e$) is usually much larger than the models used here (Burov and Diament, 1995; Hyndman et al., 2009; Tesauro et al., 2012). According to field investigations by Maerten et al. (2016), most brittle deformation can be explained using linear elastic material properties. Furthermore, the focus is not on stress changes during the co-seismic phase (e.g., Lin et al., 2013; Brodsky et al., 2020; Shi et al., 2020; Zhang and Ma, 2021), or deformation over several seismic cycles. The focus is on the quasi-static stress state in the inter-seismic
phase.

The reference geometry is a normal faulting structure with a fault dip of $60°$, but the applied boundary conditions result in a thrust- to strike-slip faulting regime at the depth, where stresses are plotted, usually at $-660$ m. Even if most models use specific structures and specific stress regime conditions, other structural settings or faulting regimes are covered by some of

the models or specific result presentations. These are the variation of the dip angle (Fig. 15 and 16), the variation of the strain (Fig. 23) and the variation of the depth for the reference model (Fig. 7). Therefore, results for all stress regimes and faulting structures are provided. However, the overall behaviour remains unchanged.

The specific objective was to investigate how faults can lead to stress rotations since this has been claimed to be the reason for observed stress rotations on scales of 10's of km. However, for most scenarios only stress magnitudes are shown here. This is of course due to the fact that many models do not show $S_{Hmax}$ rotations. Visualising the stress magnitudes gives a much broader insight into the effect of faults on the stress state. And, if the stress magnitudes change, stress rotation is possible, but if the magnitudes do not change, rotation can be ruled out. Therefore, the stress magnitude visualisation used also acts as a proxy for potential stress rotation.

To allow good comparability of modelling results, constant boundary conditions have been used, with a few exceptions. The models with different strain have different stress magnitudes as a result. For models having a different extent or a variable Young's modulus, the boundary conditions were scaled accordingly to ensure comparability. The models with a lower Young's modulus in the fault zone and low friction contact faults dissipate localised stresses, which has not been corrected, as the influence on the result are small.

## 4.2  Discontinuity approach: contact elements

Several of the model scenarios use contact elements to represent a fault within the model. This is the case for the reference model, the variation of the friction, the fault dip- and fault strike angle, the Young's modulus variation in the host rock, the model size and the boundary conditions. The overall observation is an increase of the stress components ($S_{Hmax}$, $S_{hmin}$ and $S_V$) in the footwall block and a decrease within the hanging wall block, both next to the fault (Fig. 24 a-d). In contrast, the von Mises stresses decreases in the footwall block and increases in the hanging wall block. This is the case as $S_V$ varies more than the other stress components.

For these contact surfaces, no cohesion ($C=0$) is used, which is nevertheless a reasonable and conservative simplification in particular for pre-existing faults or fault zones, as granular material have a very low cohesion: $C < 1\,\text{kPa}$ (Schellart, 2000). On the other hand, cohesion strengthening can increase the cohesion to $C > 1\,\text{MPa}$ (van den Ende and Niemeijer, 2019), $C = 8\,\text{MPa}$ (Muhuri et al., 2003) or $C = 35\,\text{MPa}$ for very high temperatures (Tenthorey and Cox, 2006). According to Tenthorey and Cox (2006), cohesion will reach 3 MPa for a 100-year earthquake recurrence interval at a depth of about 2 km.

The used friction coefficients for the contact surfaces reach from 0.1 over 0.4 (reference model) to 1.0 and larger. In the past, it was assumed, that the friction coefficient of faults is about 0.6 to 0.85 (Brace and Kohlstedt, 1980; Byerlee, 1978; Brudy et al., 1997). But the friction can be much smaller, if clay minerals dominate (Byerlee, 1978; Lockner et al., 2011), in the case of dynamic offset (Di Toro et al., 2011; Boulton et al., 2017) or for high pore pressures (Blanpied et al., 1992; Byerlee, 1993).

Low friction is also expected for large fault (zones) or subduction zones (Bird and Xianghong Kong, 1994; Carena and Moder, 2009; Iaffaldano, 2012; Fulton et al., 2013; Carpenter et al., 2015; Houston, 2015). The friction coefficient is in the order of 0.08 for the 2011 Tohoku-Oki Earthquake (Fulton et al., 2013), $\mu = 0.12$–$0.25$ or $0.05$–$0.2$ for the San Andreas Fault (Bird and Xianghong Kong, 1994; Carena and Moder, 2009) or for tremors in general $\mu = 0$ to $0.1$ (Houston, 2015). Iaffaldano

(2012) assumes a friction coefficient of 0.01 to 0.07 for large scale plate boundaries. However, the investigated range of friction cover this variation well, except for $\mu < 0.1$.

As a free surface, or a fault with very low friction coefficient, is unable to build up shear stresses (Hafner, 1951), principal stresses will be parallel and perpendicular to the surface (Hudson and Cooling, 1988; Osokina, 1988; Rawnsley et al., 1992; Petit and Mattauer, 1995; Bell, 1996; Camac and Hunt, 2009). A classic example is the San Andreas Fault (Mount and Suppe, 1987), where the interpretation of borehole breakouts and drilling induced tensile fractures from near-by borehole indicate in $S_{Hmax}$ orientations that are almost perpendicular to the fault (Zoback et al., 1987; Mount and Suppe, 1992). However, the

distance of these boreholes is $>1000\,\text{m}$ from the fault core in most cases and thus it is questionable that the derived $S_{Hmax}$ orientations can be used as an observable for the fault strength. Hickman and Zoback (2004) show in their analysis of borehole breakouts and drilling induces tensile failures of the SAFOD borehole through the San Andreas fault that significant $S_{Hmax}$ rotations can only be resolved in the near field of the fault.

### 4.3  Continuity approach: Weak elements as fault zone

#### 4.3.1  Young's modulus variation in the fault zone

Fault representation by elastic weak elements exhibits no significant stress variation pattern using three elements (Figs. 10), compared to the reference model using contact elements. Even, if the number of elements representing the fault zone is increased to nine (Figs. 11), the stress pattern is hardly different. Only close to the fault, a stress drop can be observed for $S_{Hmax}$, $S_{hmin}$ and $S_V$. The von Mises stress increases locally, as the $S_{Hmax}$ decrease is lower than for $S_{hmin}$ and $S_V$. Localised swing-in

effects can be observed, from the extent, most probably an artefact of the mesh resolution.

Fault zones are a 3-D structure consisting of the fault core the damage zone embedded within the host rock (Chester and Logan, 1986; Caine et al., 1996; Faulkner et al., 2003, 2006). Previous work suggests, that the Young's modulus of the host rock decreases towards the damage zone, where the Poisson's ratio increases in the same way (Casey, 1980; Faulkner et al., 2006; Isaacs et al., 2008). However, the variation of the Poisson's ratio is not tested here. Observed reduction of Young's

modulus is from $55.4\,\text{GPa}$ down to $16.2\,\text{GPa}$ (Isaacs et al., 2008) or a reduction of about $6.5\,\text{GPa}$, e.g., from $66\,\text{GPa}$ to $59.5\,\text{GPa}$ (Faulkner et al., 2006). The here investigated range from $E = 15\,\text{GPa}$ to $0.25\,\text{GPa}$ covers a large material property range. According Treffeisen and Henk (2020a), the amount of Young's modulus contrast have a strong impact on the resulting stress perturbation. Overall, the fault representation by means of elastically soft elements did not provide a stress pattern as the contact surface method did. It is probable that representing a fault using only elastic weak elements is a method of stress dissipation

rather than an accurate representation of low friction faults.

#### 4.3.2  Friction variation within the 3-D elements

The models having a 3-D-representation of the fault with a lateral variable number of elements, are allowed to fail according to the Mohr-Coulomb-Criteria. The resulting stress state by a friction angle of $\phi = 30°$ and a cohesion of $C = 0.1\,\text{kPa}$ did not show much difference (Fig. 13), compared to a model without a fault representation. Magnitude changes are in the order of

less than 1 MPa next to the fault zone. The models with a lower friction ($\phi = 25$, 20 and 15°) displays larger stress perturbation in the vicinity of the fault (Fig. 14). The magnitude of stress perturbation is larger for the model using a friction angle of 15°, compared to the reference model with contact surfaces. The overall pattern is complex, some of the trends are similar, but the stress magnitudes are not decoupled when crossing the fault zone. As previously discussed, a low friction can be assumed for present-day fault activity. However, resulting stress patterns differ to the results using contact elements. The continuous finite element mesh does not allow a mechanical decoupling. This may be different for other methods such as DEM where resulting behaviour depends on the number of elements and the friction (Hunt et al., 2004).

### 4.3.3 Cohesion variation within the 3-D elements

Usually, the key driver between intact rock and the fault using the Mohr-Coulomb-failure criteria is not the friction coefficient, but the cohesion. Even from the modelling perspective, cohesion has the largest impact (Treffeisen and Henk, 2020a) on the stress state. Therefore, models with elements that have elasto-plastic rheology employ the same friction ($\phi = 30°$, or $\mu = 0.58$), but a very low cohesion $C = 0.1$ kPa within the fault zone, in contrast to $C = 500$ kPa outside this area. This is also the case for elements with elasto-plastic rheology, even when the number of parallel elements reaches eight.

## 4.4 Distance of stress disturbance to faults

### 4.4.1 Far-field vs. near field

We have not specified the exact distance for the far-field or near-field, as such a distance depends on the orientation, properties, and size of the fault as well as on given stress field in the surrounding model volume. Fig. 1 and the previous content suggests, that the far-field is beyond about 100 m to the fault for intact host rock. As the ratio of displacement to fault length is about 1:100 (Torabi and Berg, 2011), even for a fault with a length of 10 km, the fault off-set can be up to 100 m. Depending on the faulting type, a limited correlation between fault displacement and thickness of a damage zone can be observed (Childs et al., 2009; Torabi and Berg, 2011). But the thickness of the damage zone is limited to a maximum of several hundred meters (Faulkner et al., 2010; Savage and Brodsky, 2011). However, for faults with a wide damage zones the impact of such a zone on the host rock is unlikely to be greater than for narrow fault zones, using the distance from the damage zone as a measure.

The impact of the different modelling approaches on the stress state differs. But a significant stress perturbation is spatially limited to a distance of maximum 1000–2000 m next to the fault. Fig. 24 provides an visual overview of modelling results. This major assumption is supported by several authors using different approaches from a more map-view perspective (Petit and Mattauer, 1995; Su and Stephansson, 1999; Provost and Houston, 2001; Yale, 2003; Faulkner et al., 2006). Also, observations from wells support that, where the stress perturbation is usually <200 m away from the fault (Stephansson and Ångman, 1986; Barton and Zoback, 1994; Brudy et al., 1997; Tamagawa and Pollard, 2008; Lin et al., 2010). A rotation of about 90° within less than 200 m in the vicinity of a fault has been observed near the Taiwan Chelungpu-fault (Lin et al., 2010) or at the Lansjärv well (Sweden, Bjarnason et al., 1989).

Only models with an oblique fault orientation relative to the maximum compression can achieve significant $S_{Hmax}$ rotation. Especially those models with a low friction ($\mu = 0.1$, Fig 19) show $S_{Hmax}$ rotation of up to 14° next to the fault. However, at a distance of 1500 m the deviation is smaller than 5°, which is quite below the uncertainties of the stress orientation indicators. Only when the friction coefficient becomes unrealistically small for faults in the inter-seismic phase (<0.1), larger rotations can be observed by the models at a distances of >1500 m away from the fault.

The relative stress state affects the spatial stress perturbation (Pollard and Segall, 1987). Therefore, Yale (2003) assumes, that in the case of low differential stress, the spatial extent of stress perturbation is able to be observed for up to several kilometres away from the fault. This fits in general to the results of the models varying the lateral strain, where the stress magnitude variation near the faults increases with a larger differential stress. Some previous models show more spacious far-field stress perturbations (Tommasi et al., 1995; Sánchez D. et al., 1999; Camac and Hunt, 2009; Maerten et al., 2002), which are most probably an artefact of a too coarse mesh resolution.

### 4.4.2 Vertical rotation of the stress tensor

Usage of the reduced stress tensor ($S_{Hmax}$, $S_{hmin}$ and $S_V$) is based on the assumption, that $S_V$ is a principal stress. However, near to a weak and non-vertical fault, the principal stress orientation will be vertically distracted, as principal stresses are always parallel to oblique to a free surface. This leads to a variation of all reduced stress components, including the shown $S_V$ magnitudes. In the case of a thrust faulting- or strike-slip regime, $S_V$ will be larger in the hanging wall block, and smaller in the footwall block, next to the fault (e.g. Fig 5). The opposite can be seen for a normal faulting regime, e.g. stress plots at greater depth (Fig. 7 at -2800 m).

### 4.5 Magnitude of stress perturbation

A decrease of horizontal stresses near the faults in the hanging wall, and an increase in the footwall is reported for the Forsmark DBT 1 well (Sweden, Stephansson and Ångman, 1986). Less borehole breakouts in the hanging wall block and more in the footwall block are observed from the KTB well (Germany, Barton and Zoback, 1994). A reduction of $\sigma_3$ by about 5 MPa has been observed within less than 10 m near a tunnel at the Grimsel test site (Switzerland, Krietsch et al., 2019). All these observations fit to the results of the models having a fault representation by contact elements, where the horizontal stresses are smaller above the fault (Fig. 25), and the horizontal differential stress is smaller in the hanging wall block (Fig. 26). The latter would make the occurrence of borehole breakouts less likely in the hanging wall.

In contrast to that, larger horizontal stresses above a fault have been observed for the Lansjärv well (Sweden, Bjarnason et al., 1989). The maximum horizontal stresses are observed about 100 m above the fault in the hanging wall block, which points also to other causes. One possible explanation is the lithological variation in that well, where several pegmatites and amphibolites in that depth range have been observed (Bjarnason et al., 1989), which eventually provide larger magnitudes as a result of a larger Young's modulus.

According to Su and Stephansson (1999) is the magnitude variation positive correlated with the stress ratio and negative correlated with the friction. This can be clearly confirmed by this study (Figs. 8 and 23), where the stress variation near fault is

largest for low friction models and models with a larger strain variation. Observation indicate that stresses decrease near a fault after an earthquake (Zhou et al., 2012; Wang et al., 2015; Li et al., 2023). This can be confirmed by the models for the hanging wall, but not for the footwall block. Either the observations are from the hanging wall block only, or other factors, like the 3-D structure, are responsible, which are not represented by the models, used here.

## 4.6 Other potential factors

All models analyse the variation of stress components and the orientation towards generic models with only one homogeneous fault. The extent to which the results can be applied to other scenarios remains questionable. There are some scenarios where we assume that other factors could have a greater influence on the stress state. These include extensive settings such as horst- and graben structures, listric faults or step-over zones. In such cases, whole blocks may be completely decoupled, either by faults or by any kind of decoupling horizon (salt, wet clay or pore over-pressure). The stress state in such a block is then dominated by gravity only. One potential example of this is the Arches National Park in Utah, USA, where the joints are almost perpendicular to the normal faults and are constant over several hundred metres (Kattenhorn et al., 2000). Secondary faulting also provides a possible explanation for the complex stress pattern within the Viking Graben (North Sea, Maerten et al., 2002). According to Siler (2023), large stress perturbations can be caused by a fault step-over structure in a hydrothermal systems over a distance of more than 1000 m in the Great Basin, western United States.

Faults or fault zones in nature are never as planar structures, as assumed by the presented models. Roughness plays a role, but the roughness in the direction of previous slip is much less, than in other directions (Power et al., 1987). The geometrical complexity are a result of non-planarity (bending, listric, bifurcation), combination or coalesce of faults (step-over- or relay zones) or others (e.g. Roche et al., 2021). Fault zones can exist out of several single parallel faults, which probably would produce a wider distributed area of stress perturbation. Pore pressure, especially above hydro-static has a significant impact on effective fault normal stresses (Blanpied et al., 1992; Byerlee, 1993). Despite the large number of models presented, such complex structures or properties have not been tested.

Stress changes near the fault tip (e.g. horsetail fault terminations) led to a complex stress pattern (Segall and Pollard, 1980; Rispoli, 1981; Homberg et al., 1997; Siler, 2023). To model that, using only linear elastic material properties would result in unrealistic local stress peaks as elastic energy would not be dissipated by plastic deformation. Therefore, such structures are not considered here. However, it can be assumed that stress changes induced by fault tips are negligible at distances of a few kilometres from the fault (Segall and Pollard, 1980; Su and Stephansson, 1999).

## 5   Conclusions

The results of our study show that the static fault friction coefficient, rock strength, stiffness and density contrast of the fault significantly affect the stress tensor beyond the fault core. However, the stress magnitudes as well as stress tensor orientation is not significantly changed beyond a distance of about 1000 m. $S_{Hmax}$ rotation is only observable when the overall orientation of $S_{Hmax}$ is oblique to the fault strike and the static friction coefficient is low (e.g. $\mu = 0.1$). From these findings we can

conclude that many of the stress tensor rotations that are documented in recent publications based on high density data sets (Heidbach et al., 2007; Pierdominici and Heidbach, 2012; Rajabi et al., 2016, 2017b; Lund Snee and Zoback, 2018, 2020) are probably not controlled by faults. Other factors probably play a greater role, like variable rock property (e.g., Reiter, 2021) or the superposition of plate boundary forces with different orientation and magnitude (Ferreira et al., 1998; Rajabi et al., 2017a). Specific fault setting could also play a roll, like decoupled graben blocks (Ferreira et al., 1998; Rajabi et al., 2017a) or secondary faults in extensional settings (Maerten et al., 2002), fault termination or transfer zones (Siler, 2023). However, it is doubtful that their far-field effect extends beyond 10 km.

**Symbols**

**Table 3.** Explanation of the symbols used

| | |
|---|---|
| $C$ | Cohesion |
| DEM | Discret Element Method |
| $E$ | Young's Modulus |
| FDM | Finite Difference Method |
| FEM | Finite Element Method |
| FVM | Finite Volume Method |
| $g$ | Gravitational acceleration |
| $S_{Hmax}$ | Maximum horizontal stress |
| $S_{hmin}$ | Minimum horizontal stress |
| $S_V$ | Vertical stress |
| X, Y, Z | Coordinates (cartesian) |
| $z$ | Depth |
| $\epsilon$ | Strain |
| $\mu$ | Static friction coefficient |
| $\nu$ | Poisson's ratio |
| $\rho$ | Density |
| $\sigma$ | Stress tensor |
| $\sigma_1$ | Largest principal stress |
| $\sigma_2$ | Intermediate principal stress |
| $\sigma_3$ | Least principal stress |
| $\sigma_D$ | Differential stress |
| $\sigma_{vM}$ | von Mises stress |
| $\phi$ | Friction angle |
| $\psi$ | Dilation angle |

*Author contributions.* KR: study set-up, model preparation, writing, discussion, OH: study set-up, discussion MZ: discussion

*Competing interests.* The contact author has declared that none of the authors has any competing interests.

*Acknowledgements.* Some of the results of that study was first presented in Heidbach and Reiter (2019). This study was partly funded by the National Cooperative for the Disposal of Radioactive Waste (Nagra), Switzerland and the Bundesgesellschaft für Endlagerung (BGE) within the project SpannEnD II (www.spannend-projekt.de).

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

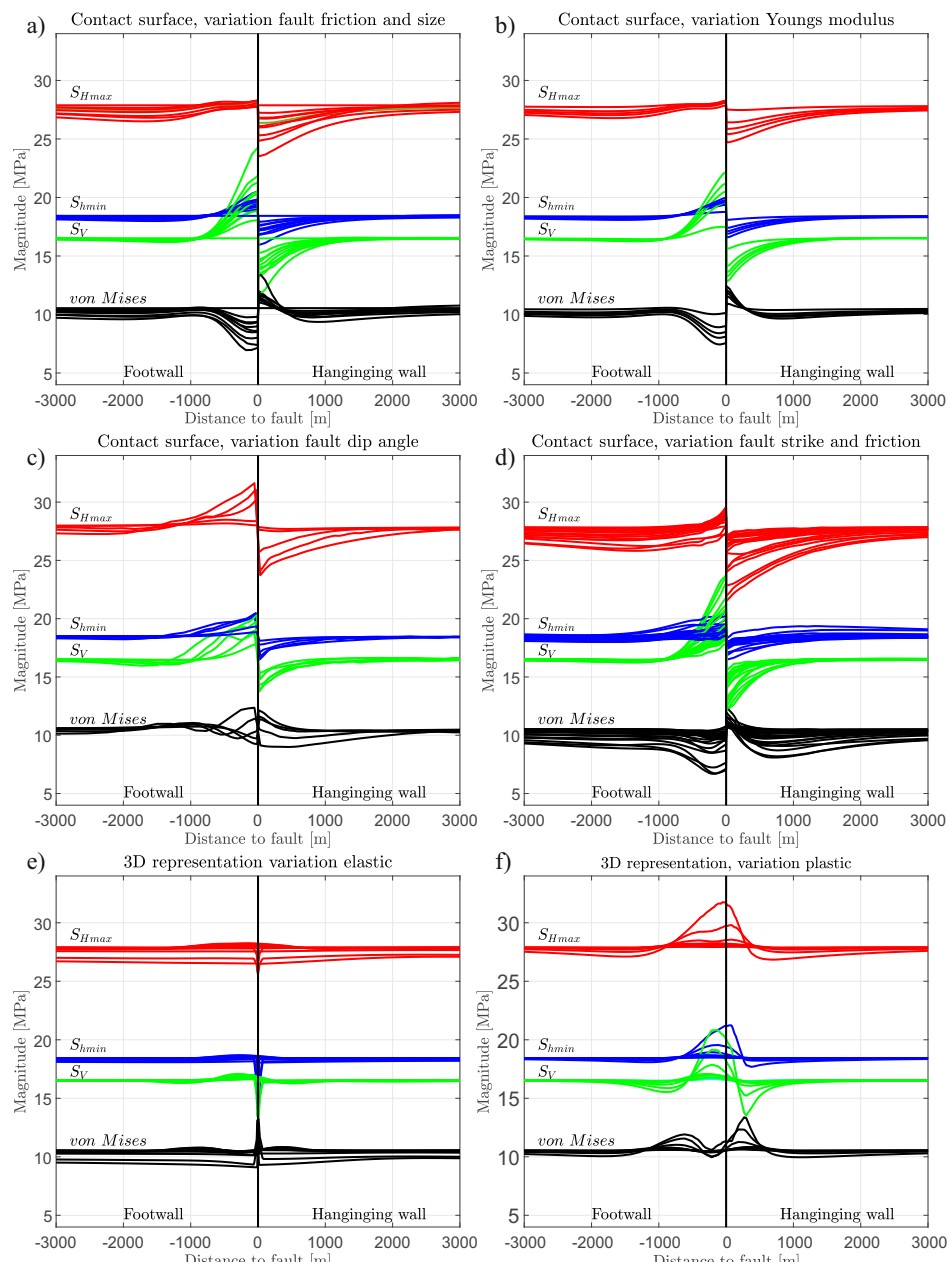

**Figure 24.** Summary illustration of the results from various presented models. Subfigure a) shows the impact of the fault friction ($\mu = 0.1$, 0.2, 0.4, 0.6, 1.0 and $> 1.0$) using contact elements (Fig. 8) and the influence of the fault size and model size (Fig. 22), b) displays the influence of a variable Young's modulus of the host rock on the stress state near and far the fault (Fig. 20). Subfigure c) shows the impact of a variable fault dip (Figs. 15 and 16), where d) illustrates the impact of a variable fault strike and additionally friction variation ($\mu = 0.1$, 0.2, 0.3 and 0.4) on the stress state resulting from a fault represented by a contact surface (Fig. 17 and 18). The impact of a fault representation by 3-D elements is shown, where e) elastically weak elements are with a different stiffness (Figs. 10 and 11) and f) where the elements are allow to plastifiy as a result of a variable low friction (Fig. 14) and a laterally variable amount of elements (Fig. 13).

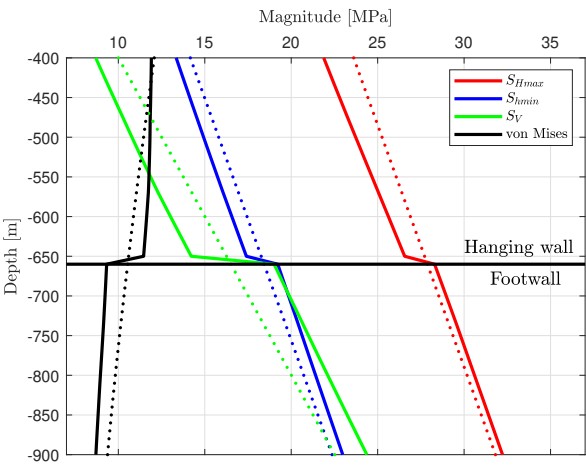

**Figure 25.** Stress magnitudes from a virtual well section for the depth range of 400 to 900 m depth, of the reference model having contact surfaces (continuous line) and a model without a fault (dotted line).

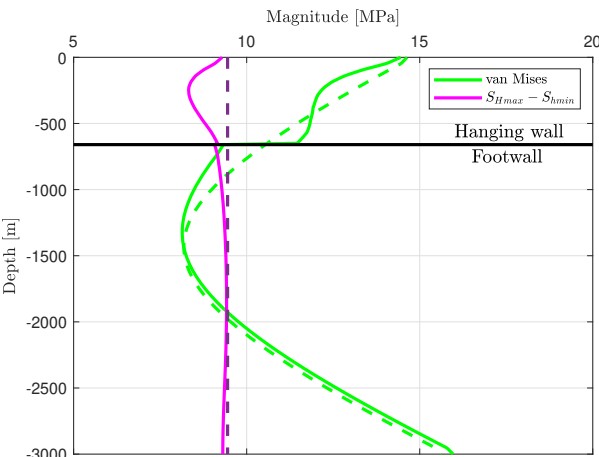

**Figure 26.** The von Mises and the difference between both horizontal stresses ($S_{Hmax}$ - $S_{hmin}$) are shown for the reference model witch contact surfaces (continuous line) and a model with a continuous mesh (dotted line).