# Peer review of "Impact of faults on the remote stress state"

_EGUsphere, 2023_

## Author Response (AR2)

**Colour explanation for Preprint egusphere-2023-1829 :**

**Reviewer comments (Sept 2023)**

**Comments from authors (3 Nov 2023)**

**Changes: line in track changes file | revised manuscript**

**Reviewer 1: Chris Morley, 24 Sep 2023**

*The paper investigates, through numerical modelling, a range of parameters associated with faults that exert an influence on stress magnitude and tensor orientation. The discussion of the problem, methodology, and presentation of the results are good, and reasonable. We are taken through a series of experiments that depart from a reference model (variable coefficient of friction, 10 thin layer, 3 weak elements; 30 m of 9 weak, elastic elements; staircase elements with elasto-plastic rheology; 4 staircase elements with elasto-plastic rheology; influence of fault dip angle on stress components; effect of strike angle on stress magnitude; Youngs modulus on stress perturbation; fault size; variable strain on stress components). This provides important information about the stress variations associated with changing a range of parameters around a single fault and provides support for conclusions from previous studies. They conclude that stress magnitudes and stress tensor orientation is not significantly affected beyond distances in of c. 1,000-1,500 m (check line 408, I assume that the . should actually be a ,). Up to this point I would accept the manuscript as it stands. It is well written, the illustrations are sufficient, and the referencing is appropriate. It is the second half of the conclusions regarding fault controls on regional stress tensor rotation where I question whether the data in the study really supports such a strong conclusion. There is very little in the experiments that addresses stress tensor rotation. I may be wrong, but I get the impression that it is assumed that variations in stress magnitude are proxies for stress tensor orientation (i.e. stress tensors will only deviate from regional in the narrow region where stress magnitudes are perturbed). There are no figures provided that show how the principal stress orientation are perturbed in the model by varying the fault strike and dip values. Figure 16, which is the only figure that address changing the strike angle of a fault, only plots variations in stress magnitude, it does not address orientation of the stress tensor. Consequently, I question why over half of the text in the conclusions focuses on stress tensor rotations, when none of the modelling data presented in the text directly addresses this topic. It is an important conclusion to address, since it affects our understanding of continental and basin-scale stress variations.*

*The experiments described in the paper focus on the effect of a single fault on stress variations. To take the results of this experiment and conclude that stress tensor rotations are not controlled by faults, but by rock properties seems a great jump. I have chosen an area in the attached figure (Kenya Rift) with an admittedly high density of faults to make the point, that you have to consider the effects of a fault population, not a single fault, if you are going to discuss basins. In the image these are quite large faults because they are visible on satellite data, the yellow line is 30 km long, and 30 faults intersect that line. Hence the fault spacing is 1 km. Hence, since this study has concluded that near field stresses are significant around 1000-1500 m from a fault, then such fault spacing should significantly influence the stress tensor orientation. The problem with trying to address faulting is that the issue is not just an individual fault core and its damage zone. Faults tend to come in large populations of*

*fractures. A large fault (kms displacement) maybe accompanied zones of secondary faults (10's-100's m displacement) that are present in zones kilometers away from the main faults (just one example are the areas of conjugate faulting that can be >5 km wide, that develop in the hangingwalls of major listric faults, and are located several kilometers into the hangingwall). We haven't even got into what fault-related joints might do to rock properties. Consequently, I would contend that a basin with its highly variable distributions of faults, at a variety of scales is a very different proposition, in terms of its potential effects on the stress tensor, to the single fault investigated in this study, and as a result the second conclusion needs to be reconsidered.                    Chris Morley*

Many thanks to Chris Morley for the thorough evaluation of the manuscript. He is right that the conclusion makes strong assumptions that exceed the results of the models. This will be considered more carefully in the final version of the manuscript. In the following several raised topics are addressed separately below:

**Rotation of the SHmax orientation at faults:**

Since almost all models, except for the model series with a variable strike of the fault, do not cause any rotation of SHmax, no visualization has been carried out so far and not to further increase the already large number of figures. However, it was stated in the text (line 162, 179, 194, 201, 216, 229, 245, 276) that no rotation, or to what extent rotations were observed (line 230). The models with a variable strike of the fault using a friction coefficient of 0.4 show only very small rotations close to the fault of a few degrees (line 230). These results were not shown, as this variation is well below the uncertainties that result from the interpretation of stress indicator such as borehole breakouts or drilling induced tensile fracture to estimate the SHmax orientation (at best 10 – 15° for high quality data). Inspired by the reviewers, we set up an additional series of models with a variable strike of the fault, using a lower friction coefficient. Even with a static friction coefficient of 0.1 (friction angle 6°), the rotation is <5° at a distance of more than 1500 m to the fault. Only when the friction coefficient becomes unrealistically small for faults in the interseismic phase (<0.1), larger rotations can be observed at distances of >1500m These additional findings will be included and discussed in the new version of the manuscript.

**Changes: 300-315 | 247- 259**

**Stress magnitudes as a proxy:**

It is correct that the variation in stress magnitude is in some instances used as a proxy for stress rotation. This can be critically discussed. However, the magnitudes of the individual components of the stress tensor can change, for example because of a fault, which results in a rotation of the principal axes of the stress. Therefore, any rotation in the stress state is necessarily associated with changes in the stress tensor components magnitudes with respect to each other. The same applies for the reduced stress tensor. These magnitude changes are not only reflected in the change in orientation of the principal stress axes but surely also in their magnitudes. Thus, it can be assumed that if the stress magnitudes do not vary, stress rotation will be neglectable.

**Changes: 392-397 | 327- 333**

**Rock Properties:**

To mention rock properties in that context seems to be a jump, but in contrast to the faults, used in the manuscript, the variation in rock properties (Young's modulus) can lead to significant largescale stress rotation (>10 km; Reiter, 2021).

**Multiple faults/fault zones**

We agree that the results obtained from individual faults in the model cannot be transferred 1:1 to complex systems, such as the Kenya Rift with several parallel faults, large listric faults or wide fault zones. Thus, this model cannot be used to estimate the impact of such fault systems. Nevertheless, it can be assumed that the far-field effect of such complex systems on neighbouring areas, depending on the overall structure, is not greater than that of individual faults. Nevertheless, and this will be pointed out in the revised version of the manuscript, there can be effects of greater range if blocks are almost decoupled, as in the case of underlying evaporites, resulting from listric faults or blocks in the centre of a graben and horst structure, cf. Kattenhorn et al. 2000. In this respect, Reviewer 1 is right that in the conclusion, assumptions based on the models have to be more carefully and differentiated.

**Changes: 537-552 | 459- 475**

**Changes: 563-575 | 482- 492**

We would like to thank Chris Morley for his contribution, to develop some models further and make the conclusion more differentiated.

**References:**

Reiter, K.: Stress rotation – impact and interaction of rock stiffness and faults, Solid Earth, 12, 1287–1307, https://doi.org/10.5194/se-12-1287-2021, 2021.

Kattenhorn, Simon A., Atilla Aydin, and David D. Pollard. Joints at high angles to normal fault strike: an explanation using 3-D numerical models of fault-perturbed stress fields, *Journal of structural Geology 22.1 (2000): 1-23.*

**Reviewer 2: Vincent Roche, 29 Sep 2023**

*General comments*

*The paper uses a numerical modelling approach to investigate the changes in stress magnitudes due to fault movement. The tested models include a cohesionless fault with various element resolutions, fault frictions, fault inclinations, strike directions, rock stiffnesses, and fault sizes, focusing on the far-field perturbation. Then, after presenting the results highlighting the effects of the different parameters, the authors discuss the model simplifications, parameterization and the impacts of other potential controls. I think this manuscript's topic is relevant to the journal, with the paper providing important general insights into stress perturbation broadening to various applications such as geothermal systems, CCS or geological disposal. The modelling design, methodology and parameters seem appropriate. The paper is well-written, but I found the structures a bit repetitive, and there are many figures. Also, I have a few main comments below and suggest publication after moderate/major revisions.*

Many thanks to Vincent Roche 2 for the thorough review of the manuscript. In the following, we will separately address and respond to individual points of the review.

*Specific comments:*

*Far-field vs. near-field: This paper focuses on the far-field stress perturbations due to faults instead of the near-field. I think such far-field is defined as a distance beyond 100 m from the fault (l.67) and is supposed to be in the intact host rock, away from the fault core and the damage zone, according to Fig. 1. Broadly, this dimension of the damage zone seems correct for a 10 km fault, but I will suggest nevertheless the authors provide some information on fault scaling supporting their chosen geometry (e.g. Childs et al., 2009, Torabi et al., 2011). Such a definition is also scale-dependent, and the far-field for a minor fault may not be the same as for a major fault. Maybe the authors could discuss their view on such a topic as well.*

This is a good suggestion to add the topic of scaling to the manuscript. We will include that subject as proposed.

**Changes: 478 - 486| 405 - 412**

*Faulting regime: The boundary conditions correspond to a horizontal shortening perpendicular to the fault's strike and extension parallel to the fault. However, according to Fig. 4, such boundary conditions result in a complex stress regime, with a thrust fault regime down to 660 m, strike-slip down to 2 km, and normal faulting below that. According to the authors, this stress state generally agrees with Northern Switzerland's. Then, the observation well for the results intersects the fault at 660 m, which is the level at which the stress regime changes from thrust to strike-slip. Therefore, it seems the result involves a rather singular transverse isotropic stress state (SV = Shmin). I will suggest that the authors explain how this impacts the results to broaden the scope of the manuscript to other stress regimes.*

It is true that a rather special stress regime prevails at the depth considered, i.e. the transition from a thrust faulting to a strike-slip regime. However, this specific situation has no influence on the results, as can be seen from the models with different displacement

boundary conditions (Figure 20). These models cover the entire spectrum of stress regimes, from a pure thrust faulting regime to a pure normal faulting regime Furthermore, now results of the reference model are plotted from different depths, representing all potential stress regimes.

**Changes: 176| 149**

**Changes: 386 - 391| 321 - 326**

***Reference model and faulting regime:*** *The reference geometry consists of a 60◦ dipping fault, which looks like a normal fault. But the normal stress regime only occurs down 2 km depth according to Fig. 4. So I think the authors should explain the rationale for using a reactivated normal fault as a reference rather than a more traditional normal fault in extension or a thrust fault in compression, for example. Maybe this is related to the context in Northern Switzerland, but I think this is worth discussing.*

**Reference model and faulting regime**

Yes, the overall model geometry is inspired by northern Switzerland. But the models analyse both, different dip angles of the faults (Fig. 14 + 15), as well as different tectonic regimes (Fig. 20). Differences are visible, but the general behaviour regarding the variation of stresses in the far field does not differ significantly.

**Changes: 386 - 391| 321 - 326**

***Stress rotation:*** *The authors investigate primarily the variation and SV, SH and Sh magnitudes, assuming those are the principal stresses. However, this is only the case if there is no rotation of the vertical stress. By contrast, if there is vertical rotation, this should induce modifications in the magnitude of SV, Sh and Sh, while the von Mises criteria should remain the same. Maybe the existence or absence of such rotations is worth discussing.*

Most of the visualisations use the representation of the magnitudes of SV, SHmax and Shmin. Close to a non-vertical structure like a low friction fault, rotation of the stress tensor will occur. Thus, close to such faults, the convention that SV is a principal stress is always assumed to be invalid, whether in the model, or in nature. The decoupling can be seen, for example, in the reference model (Fig. 5), the vertical component becomes smaller at the hanging wall block end and larger at the footwall block. However, since the effect of the fault on SHmax is significantly smaller, the von Mises stress varies. The different effects of the angle of inclination on the stress components can be seen in Figs. 14 and 15; with a low angle incidence of the fault, the variation of SHmax increases, that of SV decreases. As a result, the von Mises stress in the footwall block close to the fault increases, while that in the hanging wall block is lower. This aspect will be included in the updated discussion.

**Changes: 507 - 512| 433 - 438**

***Critical stress and failure:*** *Some models test fault friction and fault dip. However, I am unsure if this case's boundary conditions are modified. If they are not, the stress state applied to the fault may change and not always be at the same level relative to a critical state of stress. It*

*can even be greater than a critical stress state, which may be unrealistic. Maybe the authors should discuss the importance of this in their results.*

An attempt was made to keep the boundary conditions identical wherever possible. For this reason, there is also a reduction in the stress components, e.g., with low friction models (Fig. 7), or even more clearly visible for elastic elements, as a result of their width (Fig. 10), or plastic elements with low friction (Fig 13). This stress dissipation has already been mentioned in the results section (lines 173ff, 210 in the original manuscrip). An effect of this on the variation on the far field is not given. Results here do not indicate that in such a case the critical stresses could increase, it rather points in the other direction. The models are not designed for more detailed consideration of such aspects in the near field.

**Changes: 401 - 403| 336 - 337**

***Planar geometry:*** *The tested faults are planar in the models over 10 km long and 3 km deep. By contrast, faults are often complex, with bends and steps (Roche et al., 2023). Although such complexities may affect the near-field more than far-field stress, I suggest the authors discuss this point further in section 4.6, as geometry may ultimately be the main controlling factor.*

The discussion about unevenness of faults and resulting stress variations is extended. However, roughness or specific structures along the fault are not addressed by that manuscript. This can be a subject of other publications that should have detailed knowledge on a specific fault roughness.

**Changes: 537 - 548| 459 - 475**

***Figures and structures:*** *The paper has many figures (i.e. 25) and many sections. I will suggest the authors try to group some figures and results less repetitively*

**Many figures/structure**

There are certainly many illustrations in the manuscript, but this is also due to the large number of model scenarios considered. There are also some textual repetitions, as we have always endeavoured to present the models and their results with the necessary level of detail. We would like to avoid splitting the manuscript into several publications. We also have the impression that shifting some figures to the appendix would also not improve readability of the manuscript. Furthermore, we created the figures in such a way that during the final type setting of the manuscript, in the two-column style, most of the illustrations will only be limited to one column width, i.e., they will only take up half of a page width. This has less of an impact on readability than in manuscript mode plus illustrations will be right next to the according text passage.

***Technical corrections:***

Thanks to Vincent Rocher for the suggestions for corrections and additions to the literature. The corrections have been implemented, most of the literature suggestions fit well and will be adopted.

- *21-27: Stress perturbations are also important for assessing secondary fracturing near faults and associated permeability, including joint direction, secondary faulting and bed-parallel slip (e.g. Maerten et al., 2002; Kattenhorn et al., 2000; Delogkos et al., 2022).*

**Changes: 40-41 | 30-32**

*1: I think the modelling by Maerten et al., 2002 about stress perturbation is worth adding.*

**Changes: 41, 74, 545, 574 | 32, 39, 430, 466**

- *50: Photoelastic modelling has also been used to study the effects of faults on stress (de Joussineau, Soliva et al., 2010).*

**Changes: 83 | 58-59**

- *49: "Geomechnanical"*

**Changes: 75 | 51**

- *126: "1.000" Maybe use 100 0 to avoid confusion.*

**Changes: diverse**

- *132: "This shows," remove coma.*

**Changes: 147 | 137**

- *160: "decreas"*

**Changes: 212 | 164**

- *236: I will be curious to know the model's results with different Young's modulus on the HW and FW.*

**Variable Young's modulus on the HW and FW**

Resulting stress rotation will strongly depend on the friction, see Reiter (2021). Further investigations related to variable Young's modulus within the HW and FW would go beyond the intention of the manuscript.

- *See also Roche et al. (2013) for the effect of fault aspect ratio on stress perturbation.*

**Changes: reference not added**

*I hope this helps to improve the manuscript. Vincent Roche*

*References*

*Childs, C., Manzocchi, T., Walsh, J. J., Bonson, C. G., Nicol, A., & Schöpfer, M. P. (2009). A geometric model of fault zone and fault rock thickness variations. Journal of Structural Geology, 31(2), 117-127.*

**Changes: reference added**

*Delogkos, Efstratios, Vincent Roche, and John J. Walsh. "Bed-parallel slip associated with normal fault systems." Earth-Science Reviews 230 (2022): 104044.*

**Changes: reference added**

*de Joussineau, G., Petit, J. P., & Gauthier, B. D. (2003). Photoelastic and numerical investigation of stress distributions around fault models under biaxial compressive loading conditions. Tectonophysics, 363(1-2), 19-43.*

**Changes: reference added**

*Kattenhorn, S. A., Aydin, A., & Pollard, D. D. (2000). Joints at high angles to normal fault strike: an explanation using 3-D numerical models of fault-perturbed stress fields. Journal of structural Geology, 22(1), 1-23.*

**Changes: reference added**

*Maerten, L., Gillespie, P., & Pollard, D. D. (2002). Effects of local stress perturbation on secondary fault development. Journal of Structural Geology, 24(1), 145-153.*

**Changes: reference added**

*Roche, V., Homberg, C., & Rocher, M. (2013). Fault nucleation, restriction, and aspect ratio in layered sections: Quantification of the strength and stiffness roles using numerical modeling. Journal of Geophysical Research: Solid Earth, 118(8), 4446-4460.*

**Changes: reference not added**

*Roche, V., Camanni, G., Childs, C., Manzocchi, T., Walsh, J., Conneally, J., ... & Delogkos, E. (2021). Variability in the three-dimensional geometry of segmented normal fault surfaces. Earth-Science Reviews, 216, 103523.*

**Changes: reference added**

*Soliva, R., Maerten, F., Petit, J. P., & Auzias, V. (2010). Field evidences for the role of static friction on fracture orientation in extensional relays along strike-slip faults: comparison with photoelasticity and 3-D numerical modeling. Journal of Structural Geology, 32(11), 1721-1731.*

**Changes: reference not added**

*Torabi, A., & Berg, S. S. (2011). Scaling of fault attributes: A review. Marine and petroleum geology, 28(8), 1444-1460.*

**Changes: reference added**

**Reference:**

Reiter, K.: Stress rotation – impact and interaction of rock stiffness and faults, Solid Earth, 12, 1287–1307, https://doi.org/10.5194/se-12-1287-2021, 2021.